# The Current State of Knowledge on Baggio–Yoshinari Syndrome (Brazilian Lyme Disease-like Illness): Chronological Presentation of Historical and Scientific Events Observed over the Last 30 Years

**DOI:** 10.3390/pathogens11080889

**Published:** 2022-08-09

**Authors:** Natalino Hajime Yoshinari, Virginia Lucia Nazario Bonoldi, Serena Bonin, Erica Falkingham, Giusto Trevisan

**Affiliations:** 1Reumatologia, Hospital das Clinicas HCFMUSP, Faculdade de Medicina, Universidade de São Paulo, São Paulo 05508-220, Brazil; 2DSM—Department of Medical Sciences, University of Trieste, 34149 Trieste, Italy

**Keywords:** Lyme disease-like illness, tick-borne diseases, Brazilian borreliosis, *Borrelia burgdorferi* L., spirochaetales forms, Brazil

## Abstract

Baggio–Yoshinari Syndrome (BYS) is an emerging Brazilian tick-borne infectious disease that clinically mimics Lyme Disease (LD) present in the Northern Hemisphere. LD is caused by spirochetes belonging to the *Borrelia burgdorferi* sensu lato complex and transmitted by Ixodid ticks of complex *Ixodes rticinus*. On the contrary, BYS is transmitted by hard Ixodid ticks of the genera *Amblyomma*, *Rhipicephalus and Dermacentor*. In 1992, the first cases of BYS were described in patients that developed EM rash, flu-like symptoms and arthritis after tick bite episodes. Since these findings, research in BYS has been developing for more than 30 years and shows that its epidemiological, clinical and laboratorial features are different from LD. *Borrelia burgdorferi* was never isolated in Brazil. In addition, specific serologic tests have shown little positivity. Furthermore, peripheral blood analysis of patients using electron microscopy exhibited structures resembling spirochete-like microorganisms or the latent forms of spirochetes (L form or cell wall deficient bacteria). For these reasons, Brazilian zoonosis was defined as an exotic and emerging Brazilian infectious disease, transmitted by ticks not belonging to the *Ixodes ricinus* complex, caused by latent spirochetes belonging to the *B. burgdorferi* sensu lato complex with atypical morphology. The Brazilian ecosystem, combined with its ticks and reservoir biodiversity, possibly contributed to the origin of this new zoonosis, which emerged as a result of the passage of *B. burgdorferi* through exotic vectors and reservoirs.

## 1. Introduction

Attempting to identify possible Brazilian patients with Lyme disease (LD) and establishing its epidemiological, clinical, laboratorial and therapeutic aspects have been a long and difficult task. This challenging research project took more than 30 years, caused many disappointments, received criticisms and produced results that we had not foreseen throughout its development. The scientific dogmas on LD, accepted by researchers from the Northern Hemisphere, needed to be ignored while new theories emerged in Brazil. This long journey certainly could not be recounted in a conventional scientific way. Therefore, we have chosen to recap, step by step, the scientific advancement that took place, aiming to disclose the discovery of this astonishing disease [1].

## 2. Historical Background

### 2.1. The Beginning

In 1989, Dr Yoshinari completed his Postdoctoral research fellowship at Tufts School of Medicine, Boston, USA, sponsored by Dr Allen C. Steere. At that time, there was still little knowledge about Lyme Disease, and its existence was unrecognized in many countries, including Brazil. In this respect, the ticks of the *Ixodes ricinus* complex had not yet been identified in Brazil [2]. Dr Steere suggested finding possible cases of LD in the country. To kick-start the project, Dr Steere kindly offered basic research supplies, including the reagents necessary to perform ELISA and Western blotting to detect antibodies against *Borrelia burgdorferi*. In addition, he provided a culture medium (BSK II) and a culture of the *B. burgdorferi* G39/40 strain, initially isolated from *Ixodes scapularis* ticks and usually employed to perform serologic tests at Tufts School of Medicine. Positive IgM and IgG control sera of North American LD patients were also donated in order to begin the research in Brazil.

### 2.2. The Research Schedule in Brazil

The Project was approved by the Ethics Committee of the Hospital das Clínicas da Faculdade de Medicina da Universidade de São Paulo (HCFMUSP) and obtained finantial support from the Fundação de Amparo à Pesquisa do Estado de São Paulo (FAPESP). The laboratory and outpatient dispensary were set up in 1989 at HCFMUSP in order to admit patients with suspected Borreliosis. The publication of a scientific paper on LD [2] helped in raising awareness on the matter, and lectures were provided to educate physicians from different departments working at the institution.

The former group was composed by Prof. Natalino Hajime Yoshinari (clinical rheumatologist from Faculdade de Medicina of USP), Prof. Domingos Baggio (entomologist from Instituto de Biociências of USP) and Prof. Paulo Yasuda (microbiologist from Instituto de Biociências of USP).

### 2.3. The Invitation from the Brazilian Ministry of Health

In 1990, one year after setting up the laboratory at the HCFMUSP to research LD in Brazil, the group received an invitation from the National Center for Epidemiology within the Brazilian Ministry of Health to attend to patients with suspected LD coming from all over the country (Reference Center for Study of Lyme disease in Brazil). In this respect, samples of blood, cerebrospinal fluid or ticks were forwarded to the HCFMUSP, accompanied by the patients’ clinical and epidemiological data.

A brief review of LD and general recommendations on how to deliver samples to the laboratory were published in the official journals of the Brazilian Societies of Dermatology, Rheumatology, Microbiology, Tropical Medicine, Cardiology and Clinical Pathology. Additionally, all Brazilian States Health Secretaries were invited to send samples or patients to the reference unit at the HCFMUSP. During the period between 1990 and 2021, our laboratory attended to 25.128 cases referred by Brazilian physicians from all parts of the country. This enormous casuistic gave us the possibility to outline the clinical picture and behavior of serological diagnostic tests performed on cases with suspected LD.

### 2.4. The First Lyme Disease Cases in Brazil

In March of 1992, Dr Alexandre de Almeida, a physician at the Instituto de Infectologia Emilio Ribas, identified two brothers aged 10 and 12 who presented skin lesions compatible with erythema migrans (EM) following tick bites [3]. They also presented with secondary annular lesions, fever, headache, myalgia and arthralgia. The boys were from São Paulo City, but at the time, their parents were in the process of building a holiday home in Cotia County, SP, a rural area located in the Atlantic Forest. Before falling ill, the children said they had been playing with a dog that died a few days later. The area is still covered in woods which are home to both wild and domestic animals.

Dr Almeida took the advice from the HCFMUSP laboratory and performed serological diagnostic tests for *Borrelia burgdorferi* using the G39/40 strain. The immunoenzymatic assay (ELISA) showed positive IgM results in both boys, which was then confirmed by Western blot (WB), revealing the presence in both patients of at least two IgM bands. IgG class antibodies were determined to be negative both by ELISA and WB assays. The boys were treated with doxycycline for 30 days and recovered completely. Serological assays were performed a few weeks later, which provided negative results.

### 2.5. First Fieldwork Survey Conducted in Cotia County, São Paulo State

The first fieldwork survey to search for reservoir animals, which carry tick transmitters of LD and possible isolation of the etiological agent of Brazilian borreliosis, was performed in Cotia County, São Paulo State, the same place where the first cases of this zoonosis were identified. This rural area is also mentioned in scientific publications; it is located in Itapevi County and is close to both cities.

The study area consisted of an apartment building located in the southwest of Itapevi County (or Cotia County), with an area of 124.58 ha, 62.07 of which are divided into lots, and 32.80 ha consist of green areas, with secondary forests at different stages of regeneration. The climate is mesothermal, with dry winters and cool summers. Small farms with domestic animals such as horses and cows infested with ticks were observed within the area surrounding the apartment building. The study was conducted from January 1995 to June 1996. Monthly trappings were carried out for five consecutive days, always on the last week of the month [4,5,6].

Captured animals were weighed, sexed and measured. All rodents were put down for identification. Marsupials were released. Ticks were collected on animals, taxonomically classified and kept in flasks with humid paper.

Blood and internal organs from small rodents and blood from the caudal vein of marsupials were collected. Macerated ticks and animals’ blood and organs were seeded in a culture medium to allow the growth and isolation of borrelias (BSK II and other culture mediums). Part of the collected materials was preserved in liquid nitrogen or frozen for future studies.

During this period, 134 animal species were captured, of which 46.3% were Didelphimorphia (*Didelphis marsupialis* and *Mamosops incanus*), and 53.6% were Rodentia. The following nine species of Rodentia were identified: *Akodon cursor*, *Bolomys lasiurus*, *Oxymicterus hispidus*, *Oxymicterus nasutus*, *Oligoryzomys nigripes*, *Oryzomis angouya*, *Rattus norvegicus*, *Euryzygomatomys spinus and Cavia aperea*.

A total of 88 ticks were collected from the animals, most of them belonging to the Ixodidae family. Immature ticks of the *Ixodes* sp at larvae or nymph stages represented 47.7% of ticks. Adult ticks were identified as *Ixodes didelphidis* (21.6%), *Ixodes loricatus* (29.5%) and *Amblyomma cajennense* (1.2%). It is important to point out that *I.didelphidis* and *I. loricatus* do not belong to the *Ixodes ricinus* complex and do not bite humans. In contrast, there are cases of Brazilian borreliosis described in the work of Brazilian researchers, who presented erythema migrans following the bite of an *Amblyomms cajennense* tick.

Non-motile spirochete-like microorganisms were observed through dark field microscopy analysis, used to examine peripheral blood smears or to analyze samples of animal blood, organs or macerated ticks seeded in a BSK culture medium. However, isolation and culture of motile spirochetes were never achieved in the BSK II medium, even after many modifications such as the addition of different animal sera or changing its nutritional components.

These spirochete-like structures were found in 9.7% of rodents’ blood or organ culture. Only three species of rodents provided positive cultures: *Akodon cursor*, *Bolomys lasiurus* and *Oxymycterus hispidus*. In total, 13% of the *D. marsupialis* blood culture showed positivity for spirochete-like organisms, and 36.4% of adult ticks exhibited positive results.

This research conducted in Itapevi County, São Paulo State, a risk area for Brazilian borreliosis, suggested that small rodents and marsupials could be reservoirs of this zoonosis; ticks of the species *I. didelphidis* and *I. loricatus* are probably responsible for the maintenance of spirochetes among wild animals. It is important to highlight that these hard Ixodid ticks do not bite humans and cannot transmit the disease. On the contrary, *Amblyomma cajennense* is the most important tick vector in the country, responsible for causing Brazilian spotted fever, and also seems to be one of the vectors and transmitters of Brazilian LD. This study, conducted in an area at risk for LD, demonstrated for the first time how difficult it is to isolate and cultivate spiral motile spirochetes from reservoirs.

Abel et al. (2000) [7] performed the same kind of research in Cotia County, São Paulo State, and achieved similar results. Forty-four marsupials, seventy-seven rodents and one hundred sixty-one ticks were captured throughout the Atlantic Forest region. Animal blood and organs (liver and spleen) of rodents, as well as triturated ticks, were inoculated in BSK II culture medium. Twenty-one culture samples showed spirochete-like microorganisms through dark field microscopy analysis, but not the presence of the common spiral-shaped microorganisms.

An important epidemiological feature caught our attention when reviewing the clinical history of the first two boys diagnosed with LD. Since they had had contact with a dog that had died before the children became sick, we assumed the possibility of co-infection of Borreliosis and Babesiosis. In fact, after performing an ELISA and a WB test for *Babesia bovis* [8], we observed high titers of IgM antibodies in both boys. These surprising results drove us to compare the serological response of IgM and IgG class antibodies in 59 patients with Borreliosis with that of 40 healthy people. The study confirmed the existence of co-infection between Borreliosis and Babesiosis in the country’s areas at risk [9].

Naka et al. (2008) [10] also found co-infection between the etiological agent of Lyme disease-like illness and Babesiosis in children from Campo Grande County, Mato Grosso do Sul. These findings are very relevant to understanding the epidemiology of Brazilian Borreliosis because Babesiosis is transmitted by ticks of the genus *Rhipicephalus*. In this regard, Babesia *bovis* is transmitted by *R. microplus*, while *B*. *canis* is transmitted by ticks of the *R. sanguineus* species. The occurrence of this co-infection suggested for the first time the possible implication of ticks from the *Rhipicephalus* genus as vector transmitters of the Brazilian LD-like illness. Later research conducted at the HCFMUSP demonstrated that this amazing and original hypothesis was true, indicating that ticks from the genus *Rhipicephalus* also participate in Borrelia transmission in the country.

### 2.6. Second Fieldwork in Search of Borrelia sp. in Ticks from an Urban Forest Reserve in the State of Mato Grosso do Sul, Brazil

The absence of ticks of the *Ixodes ricinus* complex and difficulties in isolating and cultivating *B. burgdorfer* within risk areas for LD intrigued researchers from the HCFMUSP. We wondered if another ecosystem, different from the Atlantic Forest, where the first fieldwork had been conducted, could provide new insights.

Costa et al. (1996), who worked as a rheumatologist at Universidade Federal of Mato Grosso do Sul, located in Campo Grande, Mato Grosso do Sul State, identified the first cases of meningitis associated with LD in Brazil and also identified a total of 35 patients, including patients with EM and systemic manifestations [11,12].

In 1997, he decided to search for the etiological agent of Brazilian LD and identify the ticks and reservoir animals that could be involved in its transmission within Campo Grande County [13]. The research took place in an urban area of 48.5 ha, surrounded by woods inhabited by wild animals, including capybaras, the biggest rodent found in the country and responsible for the transmission of Brazilian Spotted Fever. This biological reserve belongs to the Fundação Universidade Federal do Mato Grosso do Sul and is located in proximity to the region known as Pantanal, which is a huge plain area surrounded by water during the rainy season.

The animals and ticks were captured and processed the same way as in Itapevi County. This phase lasted five days during July 1997.

A total of 128 ticks of the *Amblyomma* genus were collected from 5 marsupials (*Didelphis albiventris)* and 17 rodents (16 *Bolomys lasiurus* and 1 *Rattus norvegicus*). Of the ticks collected, 95 (78.9%) were in larval form, and 22 (21.1%) were nymphs; the only adult was identified as *A. cajennense*. Examination under dark-field microscopy revealed spiral-shaped spirochete-like structures, most of them with little motility, in nine cultures which had been left to seed in BSK-modified medium derived from spleens and livers of the rodents, blood of marsupials and macerate of *Amlyomma* sp nymphs. Such structures could not be identified by Giemsa’s staining procedure. PCR amplification (a procedure to identify microorganisms of genus *Borrelia*) of DNA obtained from ticks and animal cultures using primers of flagellin and 16S rRNA, as described by Barbour et al. (1996) [14], provided negative results.

This second fieldwork conducted in Campo Grande County, state of Mato Grosso do Sul, confirmed the absence of ticks of the *Ixodes ricinus* complex, which are ticks usually involved as vectors in the transmission of LD in the Northern Hemisphere. It is important to note that vectors of complex *I.ricinus* are *I. pacificus* found in the USA, *I. scapularis* also observed in the USA, *I. ricinus* present in Europe and *I. persulcatus* existing in Asia. This research carried out in Campo Grande confirmed the absence of these ticks of genus *Ixodes*, transmitters of LD in Northern Hemisphere at-risk areas in Brazil. Surprisingly, for the first time, the fieldwork implicated the possible role of ticks of the genera *Amblyomma* and *Rhipicephalus* as vectors of Brazilian borreliosis. Isolation and culture of conventional spiral and mobile *Borrelia* sp were not possible, in spite of the presence of spirochete-like structures in cultures derived from animal blood, organs and ticks. It should be specified that such few mobile bacteria-like structures were also observed in human blood cultures, especially those suspected of having Brazilian LD (Figure 1).

### 2.7. Brazilian Lyme Disease-like Patients Display Low Immune Response to B. burgdorferi

Analysis of the first Brazilian LD-like cases showed a low humoral and cellular immune response to *B. burgdorferi* when compared with North American patients at active disease stage [15,16,17,18,19,20]. Brazilian patients always exhibit a low immune response to *B. burgdorferi* at all disease stages, suggesting the presence of less immunogenic spirochete in the country.

Enzyme-linked immunosorbent assays (ELISA) performed with *B. burgdorferi* G 39/40 exhibited low serological immune response in about 35% of Brazilian LD patients in the second stage of the disease. Additionally, Western blotting assay (WB) did not fulfill the diagnostic criteria suggested by Dresslet et al. (1993) [21]. According to the authors, positive IgM and IgG immunoblotting must show two of the following bands: 18, 21, 28, 37, 41, 45, 58, 93 kDa, or five of the following molecular weight bands: 18, 21,28,30,39, 41, 45, 58, 66, 93 kDa, respectively. In contrast to these data, Brazilian patients, in general, express fewer bands and usually do not fulfil WB diagnostic criteria for LD [22].

For lack of other options, we considered a positive WB that showed the presence of two IgM or four IgG bands, regardless of the molecular weight. It is important to note that this procedure matches the initial WB interpretation provided by the Tufts School of Medicine, Boston, in 1989 [23].

Despite these difficulties, when ELISA tests (IgG plus IgM) were added to the WB results, employing the HCFMUSP methodologies, we were able to observe positivity in 65% of Brazilian cases of LD-like illness. This low rate is still statistically significant when compared with the control group consisting of healthy individuals (Table 1). Other matters must be taken into account regarding the interpretation of the ELISA test adopted in Brazil. In addition to its low sensitivity, it also shows low specificity since false positive reactions were reported in patients with other infectious and rheumatic diseases such as syphilis (IgM and IgG) and visceral leishmaniosis (IgG) and diffuse connective tissue diseases such as scleroderma (IgG), rheumatoid arthritis (IgM) and neurological autoimmune diseases [3,16,19].

At the time, we asked ourselves if the multiple laboratorial passages of the *B. burgdorferi* strain G39/40 in BSK II medium caused the loss of bacterial plasmids and antigens, causing minor ELISA and WB tests sensibility to identify Brazilian LD-like patients.

In this respect, we then performed the ELISA and WB tests using the European strains of *B. burgdorferi* sensu lato, kindly provided by Dr Arno Artur Gustav Schönberg from the Federal Institute for Risk Assessment in Germany. However, we noticed that ELISA and WB tests were conducted using antigens of the *B.garini* strain 1B29, *B.afzelii* strain 61BV1 and *B.burgdorferi* sensu strictu strain 61BV3, which provided similar results when compared with North American *B. burgdorferi* sensu stricto G39/40 strain [24]. These results confirmed the low immune response of Brazilian LD-like patients to Borrelia antigens of North American or European origins.

## 3. Reasons Why the Brazilian Lyme Disease-like Zoonosis Was Named Baggio–Yoshinari Syndrome

In 1989, the HCFMUSP laboratory proposed to identify LD cases in Brazil. Over the years, typical cases of Borreliosis with erythema migrans (EM) and multisystem clinical involvement with arthritis, carditis, neurological and ocular symptoms have been detected in the country.

However, as the research progressed in Brazil, we started to observe unexpected results. The main differences noticed between the Borreliosis found in Brazil and that of the Northern Hemisphere included the absence of the *Ixodes ricinus* complex ticks in risk areas; failure to isolate and culture the etiological agent *B. burgdorferi* in BSK II medium; low sensitivity and specificity of serological diagnostic tests (ELISA and WB) to identify suspected cases in Brazil; failure to identify flagellin and OspB genes by PCR [25].

These divergences caused great confusion and suspicion around our research, and consequently, the existence of LD in Brazil was questioned. Generally, physicians and students are accustomed to studying a given scientific matter through consultation with books and scientific papers published in Medical Journals. However, as the knowledge evolved and studies were being published in the country, many serious disagreements emerged as the results gathered in Brazil were completely different from those accepted by scientists outside this country. Were it not for the respectability of the FMUSP as an institution and its researchers, the project of identifying Brazilian LD-like illness would have ceased prematurely.

Due to the findings that followed these contradictory results, which conflicted with the dogmas embraced by outside researchers on Lyme disease, our research data were never accepted for publication by International Journals. Reviewers ridiculed the possibility that Brazilian Borreliosis could be caused by ticks not belonging to the *Ixodes ricinus* complex, did not acknowledge patients who did not fulfill the serological diagnostic criteria adopted by the Centers for Disease Control and Prevention (CDC) as well as the description of spirochete-like microorganisms; the absence of molecular biology positive cases, even after performing PCR tests with primers to identify the *Borrelia* genus; reports of relapsing cases, even after antibiotic use as recommended by the CDC criteria. Therefore, at the time, we asked ourselves what kind of disease was occurring in Brazil and how to confront it [26,27,28].

As a consequence of these issues over time, this unusual tick-borne disease was named in many different ways, such as Brazilian Lyme disease, Brazilian Lyme disease-like illness, Lyme disease-simile illness, Infectious-reactive Lyme disease-like syndrome; and finally, Baggio–Yoshinari Syndrome (BYS) [29].

The term Baggio–Yoshinari Syndrome (BYS), adopted in 2005, expresses a tribute to Prof Domingos Baggio, who died a few years after the beginning of our project, and also implies that we have a new Brazilian tick-borne disease, completely different from LD, despite the many clinical similarities with Lyme Borreliosis. In this regard, it deserves the attention of specific research to understand its etiological agent, its epidemiological transmission cycle, laboratory diagnosis, clinical manifestations and treatment. In fact, what we wish we could have stated, was that Lyme disease does not exist in Brazil, except for imported cases, and to stop making comparisons with the disease we were used to seeing in the Northern Hemisphere. The choice of adopting the eponym Baggio–Yoshinari was not an act of self-referential. What it means is that all criticism and scientific mistakes must be attributed to both researchers.

At the beginning of our studies, we thought that Brazilian Borreliosis could be similar to a disease known as STARI (Southern Tick Associated Rash Illness) [30], identified in the south of the USA, characterized by the presence of a rash similar to the EM, without the appearance of systemic symptoms. It is caused by the spirochete *B. lonestari*, isolated and cultured only in tick cells [31]. BYS differs from STARI as Brazilian borreliosis causes systemic manifestations, and its etiological agent has remained uncultivable until the present day.

Currently, Brazilian physicians accept that BYS is an original clinical and laboratorial expression of *B. burgdorferi* infection, very distinct from that exhibited by Lyme disease. This important scientific knowledge is possibly related to the passage of spirochete through different ticks present in Northern Hemisphere and in Brazil, reflecting the influence of ecology and tick biodiversity on disease expression. Due to unknown reasons, possibly, *B. burgdorferi* infecting Brazilian vectors and reservoirs, present genotypic changes (mutations?), originating morphologically and antigenically modified spirochetes, adapted to survive in our geographical and biodiversity conditions.

According to our hypothesis, studies on BYS could help to explain why LD is not clearly reported in other regions of the Southern Hemisphere (Australia, Africa, New Zealand and South America), geographical areas with important biodiversity on ticks and reservoir distribution. A preliminary literature review indicated the absence of ticks from the complex *Ixodes ricinus* in these regions, suggesting that different ticks which do not belong to the genus *Ixodes*, similarly to what occurs in Brazil, could originate from exotic Borreliosis, different from typical LD found in USA, Europe and Asia. In this respect, research on Brazilian borreliosis could help to understand the many laboratorial and clinical expressions of spirochetosis caused by *B. burgdorferi*, according to tick and reservoir biodiversity found around the world, mainly in the Southern Hemisphere.

## 4. Searching for the Etiological Agent of Baggio–Yoshinari Syndrome

### 4.1. Molecular Biology Researches

In 2000 [25], for the first time, Barros performed polymerase chain reaction (PCR) tests in DNA extracted from the blood samples of 34 BYS patients, according to procedures recommended by Liebling et al. (1993) (identify OspA) [32], Kawabata et al. (1994) (flagellin) [33] and Barbour et al. (1996) (flagellin and RNAr) [14]. All the tests were negative.

In 2008, Elenice Mantovani, for her PhD dissertation at the HCFMUSP [34], was invited to attempt to identify the etiological agent of BYS employing PCR procedures and try to understand the significance of spirochete-like structures present in patients with SBY. It is important to note that such structures are also present in normal individuals, although this finding is statistically less frequent in healthy people.

At first, 68 BYS patients were selected (group A), and blood from these patients was collected in tubes with EDTA between May 2005 and January 2007. Of these patients, 24 (35%) had shown previous erythema migrans (EM), 5 (7.3%) had arthritis, 8 (11.7%) exhibited carditis and 20 (29.4%) presented neuropathies. Positive serology for *B.*
*burgdorferi* according to HCFMUSP criteria was observed in 44 cases (64.7%).

After DNA extraction from 68 patients and from 50 normal controls, PCR tests were performed employing primers to identify: *Borrelia* spp. (gene FlaB) (Stromdahl et al., 2003) [35]; *Borrelia* spp. (gene 16 S rRNA) (Rich et al., 2001) [36]; plasmids Ip 25, Ip 28-1, cp32-4 and cp32-2/7 (Iyer et al., 2003) [37]. All PCR tests came back negative or provided doubtful results when primers to detect *Borrelia* plasmids were used.

As the first attempt to identify *Borrelia* spp. in BYS patients failed, another PCR assay was planned, selecting only 10 patients with active disease and presence of EM. All patients, six female and four male, showed some clinical manifestations such as fever, flu-like symptoms, arthritis, arthralgia and headache. Five patients reported a tick bite, and all had visited areas at risk for BYS. Five patients were classified at the acute stage of the disease (less than 3 months following the tick bite), and the other five were in the latent infectious stage, ranging from 105 to 1440 days after the tick bite.

Interestingly, one female patient included in the group presented with a chronically expanding erythematous skin lesion following a tick bite which had taken place 4 years earlier. A skin biopsy was also employed for PCR testing and histological analysis, which provided results which were compatible with localized systemic sclerosis (morphea) or acrodermatitis chronica atrophicans (ACA). She had been treated many times with antibiotics, but the skin lesion had continued to grow (Figure 2).

After DNA extraction from the blood of nine patients and from one piece of skin obtained from biopsy, we observed the amplification of the conserved gene that synthesizes the flagellar hook (*flgE*) of *B. burgdorferi* [38] provides evidence for the existence of Brazilian Borreliosis (BYS). Six out of ten cases (60%) showed positive PCR. Interestingly two patients presented long intervals between the tick bite and a positive PCR. One man who presented an EM following a tick bite in the Amazon Forest was treated with antibiotics and improved. However, six months later, he presented with knee arthritis, a PCR test was then performed, showing a positive result. The woman with the expanding skin lesion, compatible with systemic localized sclerosis (morphea) or ACA, which she had for over 4 years, was another case of persisting *Borrelia* in BYS (Figure 2).

The gene *flgE* was amplified from these patients, and upon sequencing, these positive cases revealed 99% homology to *B. burgdorferi flgE* [39]. For the first time in Brazil, we obtained a PCR test to identify BYS patients. Additionally, these PCR tests performed with *flgE* primer allowed us to demonstrate the long-term persistence of *Borrelia* in BYS, justifying the occurrence of relapsing symptoms associated with this syndrome.

Confirming the clinical relevance of the amplification of the *flgE* gene to diagnose BYS, Lopes et al. (2017) identified four patients from the Central–Western region of Brazil [40].

In addition, Mantovani (2008) [34,39] carried out the same PCR test with *flgE* primer in 47 samples of ticks collected from risk areas in Espirito Santo State, 17 being *Rhipicephalus microplus* and 30 *Rhipicephalus sanguineus*. In addition to these ticks, she included DNA extracted from the blood of 27 buffalos and 26 horses, kindly provided by Prof Adivaldo H. Fonseca from the Universidade Federal Rural of Rio de Janeiro. The PCR test was positive in two tick samples (one was *R. sanguineus* and the other *R. microplus*), in one horse and one buffalo. The positive control for all PCR tests was DNA extracted from *B. garinii* (Figure 3).

The PCR test performed with *flgE* primer showed for the first time that the causative etiological agent of BYS was the spirochete *B. burgdorferi* sensu stricto. Furthermore, this important discovery confirmed the involvement of hard Ixodid ticks not belonging to the *Ixodes ricinus* complex as tick transmitters of Brazilian Borreliosis. It also suggested the role of domestic animals as reservoirs and participants in the transmission cycle of BYS.

The reason why we chose the primer derived from the flagellar hook (*flgE*) to attempt to detect *B. bugdorferi* in BYS deserves some observation. The flagella apparatus is made up of three parts: the basal body, the hook (synthesized by the *flgE* gene) and the flagella filament (synthesized by FlaA and FlaB genes) [41,42,43] (Figure 4). Results from previous PCR tests suggested the suppression or absence of FlaA and FlaB genes in the *Borrelia* occurring in Brazil, justifying the lack of flagella and the difficulties in isolating and identifying motile spirochetes in reservoirs and patients with BYS. Otherwise, WB analysis of BYS patients always showed the presence of the 41 kDa protein. We assumed that *Borrelia* infecting vertebrate and invertebrate hosts in Brazil had possibly lost the genes responsible for filament synthesis but had likely maintained the conservative genes responsible for the synthesis of other flagella apparatus components, such as the flagella hook structure.

The *flgE* gene is formed by 1119 nucleotides. After the sequencing of PCR products obtained from PCR testing executed with *flgE* primer, the alignment of the nucleotides sequence showed a difference in only two DNA bases. Notably, this alignment was observed in BYS patients, ticks, buffalos and horses, indicating that this genotypic *B. burgdorferi* sensu stricto circulates among vertebrate and invertebrate hosts in Brazil (Figure 5 and Figure 6, Table 2). The research carried out at the HCFMUSP enabled us to understand the transmission cycle of BYS, pointing out the relevance of exotic ticks of the genera *Amblyomma* and *Rhipicephalus* and the involvement of domestic animals.

Gonçalves et al. (2013) [44] also contributed to discovering the etiological agent of BYS. The group collected ticks from horses used for animal traction in the rural area of Jataizinho, Paraná State, to search for the DNA of *B. burgdorferi* sensu lato. A total of 224 ticks were collected between February and June 2008; 75% were identified as *Dermacentor nitens* and 25% as *Amblyomma cajennense*. To amplify the *B. burgdorferi* sensu lato DNA, the intergenic space regions (ISR) between the *5S (rrf)*–*23S (rrl)* rRNA genes were used as targets for nested-PCR. Two ticks of the *D.nitens* species were positive for *B. burgdorferi* sensu lato, and the sequencing revealed 99.9% homology to *B. burgdorferi* sensu stricto.

Gonçalves et al. (2015) [45], employing the same primer, identified spirochetes belonging to species *B. garinii* and *B.burgdorferi* sensu stricto in humans in a rural area of Paraná State, Brazil

Talhari et al. (2010) [46] also contributed to discovering the presence of *B. burgdorferi* in Brazil. They found spirochetes in skin biopsies of EM in a patient from the Brazilian Amazon Forest, using specific immunohistochemistry and focus floating microscopy for *B. burgdorferi*. However, the authors failed to isolate and culture spirochetes in BSK II medium. Furthermore, they did not succeed in identifying *B. burgdorferi* through DNA extraction from skin biopsies and nested PCRs targeting conserved genes.

### 4.2. Microbiological Research

Spirochetes of the *B. burgdorferi* sensu lato complex are microaerobic, slow-growing Gram-negatives microorganisms with a spiral shape and corkscrew-like motion. They have 7 to 11 periplasmic flagella, and the protoplasmic cylinder is surrounded by an outer membrane rich in lipoproteins (Osps). The Borrelia genome is composed of a single linear chromosome and a variety of linear and circular chromosomes [41].

Although *B. burgdorferi* is difficult to culture, BSK II (Barbour–Stöenner–Kelly culture medium) is usually employed to isolate spirochete from blood, cerebrospinal fluid and skin biopsies obtained from EM rash. This time-consuming and not very productive procedure is rarely used as a laboratory diagnostic assay to detect LD infection.

Possibly, because of its structural characteristics, spirochetes were never isolated and cultured from BYS patients, ticks or animal reservoirs. Many modifications of the BSK II medium were attempted to make the medium rich enough to encourage the growth of microorganisms. Despite this, we never succeeded.

However, we observed that, a few days after seeding biological samples in BSK II medium, few motile spirochete-like organisms were observed through dark field microscopy analysis. After many attempts to try to convert non-motile structures into motile spirochetes, we discovered that the medium named SP-4 (*Spiroplasma mirum* base) (Tully, 1995) [47] was capable of expanding and maintaining these organisms for long periods of time (Figure 1).

Prof. Maria de Lourdes Higuchi from the Department of Pathology of the Instituto do Coração da FMUSP analyzed these structures under electron microscopy and observed the presence of organisms resembling mycoplasma, round bodies suggesting chlamydia and elongated bacteria-like structures (Figure 7). We carried out serological assays to detect antibodies for *Mycoplasma pneumoniae* (ELISA), indirect immunofluorescence assays (IFI) to identify Chlamydia and PCR tests (Messmer et al., 1977; Smith et al., 2004) [48,49] for both microorganisms in BYS patients. All test results came back negative, indicating that the spirochete-like structures which had been detected in BYS patients, ticks and reservoirs were not *Mycoplasma* or *Chlamydia*.

The medical literature reported that spirochetes in unfavorable conditions, such as changes within the host, pH or temperature modifications and the introduction of antibiotics into the growth medium, can provide the origin of morphologically different structures resembling round and dense Chlamydia-like structures, cell wall-deficient Mycoplasma, such as in organisms, and non-motile spirochete-like filaments [50,51] (Figure 8).

In 2010, Mantovani [34] also studied the possibility that spirochete-like structures observed in BYS patients could infect endothelial cell cultures in vitro. Initially, she seeded 1 mL of blood from the BYS patient in SP-4 medium. After three days, an aliquot of 1 mL of this suspension containing spirochete-like structures was added to two culture flasks containing adhered endothelial cells. After 10 days of incubation, endothelial cells were removed, washed and examined by electron microscopy. Spirochete-like structures were observed in one of these cultures (Figure 9).

One other experiment conducted at the HCFMUSP, in which the *B. burgdorferi* G 39/40 strain was maintained in BSKII medium culture and then introduced in co-culture with endothelial cells, showed, after 7 days, the presence of spirochete-like structures inside the endothelial cells (Figure 10).

In 2009, a curious and analogous event took place [53]. Our research group was called to investigate the presence of low motile spirochetal microorganisms with different morphologies and sizes through dark field microscopy in the blood of animals from the USP Medical School Vivarium. The bacteria did not show any sign of growth in common culture media, producing faint staining with Giemsa and silver-derived stains and negative serological and molecular tests for *Borrelia* spp. and *Leptospira* spp. Electron microscopy revealed the presence of microorganisms with Mycoplasma-like morphology. Such structures were also observed in 15 out of 26 (57.6%) employees of the Vivarium of FMUSP; however, these individuals’ clinical and laboratory exams were normal. It can be assumed that such structures were L- form (cell wall deficient) bacteria from contaminants present in the environment or derived from endogenous microorganisms present in the normal saprophytic flora.

Spirochete-like structures were present in more than 90% of BYS patients and nearly 20% of healthy individuals. Additionally, BYS patients showed the presence of round bodies resembling Chlamydia, which were absent in the blood of the Vivarium animals and workers.

We assumed that BYS could be a tick-borne infectious disease caused by *B. burgdorferi* in its altered morphologies, including L -form microorganisms resembling *Mycoplasma*, round structures similar to *Chlamydia* and non-motile bacteria which had lost their flagella.

In our opinion, these camouflaged *Borrelia burgdorferi* sensu stricto, exhibiting pleomorphic atypical morphologies, possibly appeared in our country as an adaptive process to survive the conditions of Brazil’s ecosystem. This smart adaptation by Borrelia to ticks and reservoirs present in the country explains why BYS patients express low immune response against the spirochetes and justifies many of the clinical symptoms observed in Brazil, such as the incidence of recurring episodes, frequent reactivations (autoimmunity and chronic fatigue symptoms) and the need for extended periods of antibiotic therapy.

Supporting our studies, Meriläinen et al. (2015) [54] demonstrated the presence of pleomorphic morphological forms of *B. burgdorferi* sensu lato, induced by different culture conditions originating from the presence of the spherical round bodies that keep their flexible cell envelope intact. These structures should be considered clinically relevant and deserving of novel diagnostic methods and treatment protocols.

Rudenko et al. (2019) [55] also commented that the survival of spirochetes from the *B.burgdorferi* sensu lato complex in a hostile environment is achieved by the regulation of different gene expressions in response to changes in temperature, nutrients, hosts or vectors. These altered conditions can cause changes in gene expression, providing an origin to persisters or dormant cell subpopulations which require low metabolic activity, permitting long periods without replications. According to the authors, persisters are elusive, present in low numbers, morphologically heterogeneous, multi-drug-tolerant cells that can change with the environment. Persisters can adopt varying sizes and shapes, changing their morphologies. They are capable of forming round bodies, L- form bacteria, micro colonies or biofilms-like aggregates, which remarkably change the response of Borrelia to hostile environments. Persisters remain viable and are able to reversibly convert into motile forms in a favorable growth environment, and are antibiotic tolerant, justifying recurrent disease episodes.

Karvonen et al. (2021) [56] demonstrated that by culturing *Borrelia burgdorferi* in two lines of human cell lines (chondrosarcoma and dermal fibroblasts), the presence of polymorphic structures such as blebs, round bodies and damaged spirochetes, showed cell viability after 9 days of culture. The authors suggest that the invasion of non-phagocytic cells and the lack of cytopathic effects on the host cell by *B. burgdorferi* indicates one mechanism of immune evasion for the bacteria and also explains the ability of the bacterium to adapt to different environments, as well as a strategy for persistence inside a host.

Our experiments, reinforced by recent research published in the medical literature, support the theory that *B. burgdorferi* found in Brazil, infecting patients, ticks and reservoirs, can be found at atypical morphologies. Furthermore, these non-motile and persisting structures survive mainly inside cells, eliciting a low immune response and, in consequence, enabling the spirochetes to become more resistant to antibodies and antibiotics. These particularities viewed in BYS patients are possibly a consequence of spirochete adaptation to the ecosystem and biodiversity found in the country, mainly due to the necessity of *B. burgdorferi* adaptation to survive and infect exotic Brazilian ticks belonging to the genera *Amblyomma*, *Rhipicephalus and Dermacentor.*

## 5. Differences in B. burgdorferi Antigens Expression in BYS and LD Patients

In 2000, due to controversies regarding WB expression found in BYS and LD patients, we asked Dr Roberta Gonçalves Marangoni, a PhD student at the HCFMUP, to travel to Dr Allen C. Steere’s laboratory to try and dispel the doubts on this matter. She analyzed the pattern of antibody expression in 36 BYS patients using the MarDx *Borrelia burgdorferi* (IgM and IgG) Strip Test System. A comparison was made with 20 sera samples for IgM analysis and 30 for IgG, yielded by Dr Steere.

Brazilian patients (n = 36) presented the following symptoms: EM in 16 cases (44.4%), fever in 16 (44.4%), myalgia in 12 (33.3%), headache in 6 (16.6%), fatigue in 6 (16.6%), arthritis in 9 (25.0%), neurological manifestations in 12 patients (33.0%) and cardiac involvement in 3 cases (8.0%).

BYS patients showed only a few bands for both IgM and IgG antibodies. For comparison purposes, we displayed only the IgG class of antibodies on the WB in Figure 11. None of the 36 Brazilian patients met the requirements established by the CDC diagnostic criteria for IgG WB, and only 1 BYS patient exhibited the same pattern observed in North American patients.

In BYS cases, the 41 kDa band was the most common on both the IgM (6 cases, 16.6%) and IgG immunoblotting (26 cases, 72.2%). As previously commented, *B. burgdorferi* implemented adaptation mechanisms to survive in the Brazilian environment, such as possibly losing the flagellin filament synthesized by genes Fla A and FlaB, which led to the loss of its helical movement; however, it preserved its hook structure, which justifies the presence of the 41 kDa spirochete protein.

Non-specific bands such as 10 kDa (21 cases, 58.3%), 29 kDa (3 cases, 8.3%) and 32 kDa (3 cases, 8.3%) were observed on the IgM WB of BYS patients; the same was found for IgG blotting, which showed the presence of 61 kDa (10 cases, 27.7%) and 70 kDa (5 cases, 13.8%). These two last bands are related to heat-shock proteins.

The Western blotting results of BYS patients confirmed the ELISA findings, indicating that *B. burgdorferi* circulating in Brazil gives rise to the low immune response in vertebrate and invertebrate hosts when compared with the forms of *Borrelia* found in the Northern Hemisphere. WB analysis also suggested that *B. burgdorferi* adapted to Brazilian biodiversity and had lost or suppressed many genes, making it less immunogenic, meaning less capable of producing antibodies. Gene loss or suppression can justify the presence of pleomorphic *Borrelia* in the country, such as non-motile spirochetes and round bodies resembling Chlamydia and L-form bacteria (cell wall-deficient). Spirochetes of atypical morphologies can survive inside cells, such as endothelial, mononuclear and fibroblasts, leading to chronic and recurrent infections, making spirochetes resistant to antibiotics and the host’s immune system. According to Palmer et al. (2016) [57], antigenic variation is a strategy used by a broad diversity of microbial pathogens, which include *Borrelia*, allowing them to persist within mammalian hosts, and then avoid host immunity. Chaconas et al. (2020) [58] state that one of the many fascinating features of *B. burgdorferi* is an elaborate system of antigen variation, whereby the sequence of the surface-bound lipoprotein VlsE is continually modified through segmental gene conversion events, enabling spirochetal persisting infection.

## 6. Borreliae Lyme Group

In the Borrelie Lyme Group, we can distinguish the following subgroups (Table 3) [59]:○Classical Organotropic form, with EM and transmitted by *Ixodes* sp.;○Spirochaetemica Group (*Borrelia mayonii*) with EM transmitted by *Ixodes* sp.;○Baggio–Yoshinari forms with EM, systemic manifestations, autoimmunity and chronic fatigue symptoms, with low immune response to *B. burgdorferi* and transmitted by ticks of the genera *Amblyomma*, *Rhipicephalus (R. sanguineus*, *R. bovis) and Dermacentor.*

BYS deserves a separate classification in the Lyme group due to its distinguishing epidemiological, clinical and laboratorial characteristics: (1) it is transmitted by ticks not belonging to the genus *Ixodes*; (2) the etiological agent *B. borrelia* is not isolated or cultured in BSK II-modified medium, and currently, only identified by PCR procedures; (3) spirochete, presenting typical, cork screw morphology with spiraled movement found in LD patients, was never determined in BYS patients. Brazilian researchers suggest that spirochetes in BYS exist in atypical round-shaped forms or bacteria-like morphologies; (4) *B. burgdorferi* in BYS patients gives rise to low immune humoral and cellular responses. Antibody titers are low and disappear fast in the serum of patients; (5) BYS patients present all the clinical aspects displayed by LD; however, some divergences are noted, such as the high frequency of clinical recurrences, the necessity of prolonged antibiotic therapy administered at disseminated disease stages and high risk of emergence of autoimmune and chronic fatigue symptoms. These manifestations are also observed in LD, but episodes of recurrence, autoimmunity and chronic fatigue are less significant and frequent. Particularities of *B. burgdorferi* infection in BYS, such as the presence of persisting intracellular microorganisms at atypical morphologies and them becoming more resistant to antibodies and antibiotics, could explain the propensity towards the development of reactive symptoms (autoimmunity and chronic fatigue symptoms).

## 7. Clinical Involvement of Baggio–Yoshinari Syndrome

The clinical aspects of BYS are very similar to those presented by LD in the USA and Europe. Despite the many epidemiological and laboratorial differences observed between the two diseases: (1) *Borrelia burgdorferi* in Brazil is pleomorphic in morphology and identified by PCR procedures; (2) this Borrelias strain has not yet been isolated or cultured in BSK II medium; (3) tick transmitters belong to the *Amblyomma*, *Rhipicephalus*, *Dermacentor* genera; (4) small rodents, marsupials and domestic animals act as reservoirs; (5) serologic diagnostic criteria adopted in the USA fail in Brazil; (6) in BYS there is a higher rate of clinical relapsing episodes; (7) there is a tendency for BYS patients to develop autoimmune or chronic fatigue syndrome symptoms; (8) there is a necessity for longer antibiotic therapy at the disseminated disease stage in Brazilian Borreliosis. [11,12,16,18,19,23,26,27,28,60,61,62,63,64,65,66,67,68,69,70,71,72,73,74,75,76,77,78,79,80,81,82,83,84,85,86]

Currently, Baggio–Yoshinari syndrome is defined as an emerging tick-borne disease and, up until now, it has been categorized as being specific to Brazilian territory. It is caused by *B. burgdorferi* sensu stricto, is found in pleomorphic morphologies, is non-isolated nor cultivated yet and is transmitted by hard Ixodid ticks that do not belong to the *Ixodes ricinus* complex, which reproduces a clinical picture that resembles Lyme disease, except for its tendency to present autoimmune and chronic fatigue syndrome symptoms.

Serological diagnosis of BYS is difficult and causes a lot of confusion among Brazilian physicians because BYS patients present an inadequate immune response against *B. burgdorferi* antigens of North American or European origin. The ELISA assay conducted in BYS patients shows the occurrence of low and transient antibodies to *B. burgdorferi*. Similarly, the WB assay displays the presence of a small number of IgM and IgG bands, stopping us from performing serologic diagnostics for Borrelia infection. For these reasons, serologic tests to identify antibodies to *B. burgdorferi* fail to diagnose Borrelia infection in BYS patients. ELISA and WB assays carried out in Brazilian patients, in general, do not fulfill the CDC diagnostic criteria for serologic diagnostics regarding *B. burgdorferi* infection.

Despite these problems, we conducted serologic ELISA and WB tests in BYS patients, employing some modified strategies. The ELISA test was performed as usually conducted in the USA, employing antigens of *B. burgdorferi* of the North American strain G 39/40. In contrast, due to the low sensitivity and specificity of the WB assay carried out in BYS patients, researchers of the HCFMUSP determined positive WB results and the presence of at least four IgG or two IgM bands, independent of the molecular weight of these antigens. Using ELISA plus modified WB, we identified around 65% of BYS suspected patients. This value is statistically significant compared with the positivity observed in normal individuals.

Once more, it is important to note that the use of these serologic procedures adopted in Brazil presents serious limitations, and attention should be brought to their interpretation. Unfortunately, this is the only laboratorial test available in the country. We assume that the low sensitivity and specificity of serological tests to identify *B. burgdorferi* infection in BYS patients is a consequence of the existence of low antigenic spirochete circulating in the country, adapted to survive in our biodiversity. In this respect, isolated serological tests do not permit the diagnosis of BYS patients in Brazil. Therefore, clinical and epidemiological analyses are essential tools to recognize Borreliosis in the country.

Despite all cited limitations, performing an ELISA assay in the cerebrospinal fluid (CSF) can be helpful in patients with neuroborreliosis. This immunological test, added to cytological, biochemical analysis and exams to exclude infectious agents, can help to identify cases of neuroborreliosis. Finding oligoclonal bands in CSF analysis of neuroborreliosis patients is not unusual. Findings from synovial fluid analysis and biopsy of synovial tissue performed in BYS patients are similar to those observed in rheumatoid arthritis.

In the USA, attempting to identify *B. burgdorferi* from blood culture, CSF and tissue samples in BSK II medium is considered a time-consuming and not very effective procedure. As already mentioned, the *Borrelia* found in Brazil, which is characterized by the above-mentioned atypical morphology, has never been isolated or cultured. PCR tests are only requested at the disseminated stage of the disease or when the persistence of chronic expanding skin lesions is present.

In conclusion, the identification of BYS patients is not an easy task. When the typical and distinguishing EM at the site of the tick bite is reported, the occurrence of other systemic symptoms deserves a thorough anamnesis in order to exclude other conditions that could mimic the manifestations exhibited by BYS patients. In this respect, current and past epidemiological events, such as reports of one or more tick bites, contact with animals or trips to risk areas, are of critical relevance to help in the identification of BYS cases.

On top of these considerations, it is important to keep in mind that BYS continues to progress, especially if not recognized and treated at the early stage of the disease. Months, years or perhaps decades may elapse between the initial infection and the development of current symptoms. It is important to remark that the diagnosis of BYS is based on clinical and epidemiological data while ruling out other conditions.

### 7.1. Clinical and Progression of BYS

Based on the analysis of its evolution and clinical manifestations, Baggio–Yoshinari Syndrome presents two main stages. The initial phase, or infectious stage, starts after the tick bite and inoculation of *B. burgdorferi* in the skin (localized stage), which is followed by dissemination. During this stage, the spirochetes move to the different organs through the vascular or lymphatic circulation systems. On most occasions, the infection improves spontaneously. Sometimes the microorganisms cause inflammatory processes and damage to many organs or tissues such as the joints, the heart, the peripheral and central nervous system, the eyes and the skin (secondary stage). Eventually, spirochetes may remain dormant, and the infection can linger in a latent stage for months or even years. However, at any moment, it can re-activate, causing the disease to progress into the tertiary stage of infection.

It is common that BYS patients in Brazil, especially those who have not been treated with antibiotics or have undergone only a few days of treatment, can show, even months or years later, laboratory manifestations or multisystem symptoms compatible with autoimmune or chronic fatigue/myalgic encephalomyelitis symptoms (CFS/ME). This phase is known as the reactive stage of BYS, which is possibly triggered by immune activation in response to spirochetal antigens.

In the Northern Hemisphere, some patients treated for LD with antibiotics present non-specific symptoms such as fatigue, sleep disorders, arthralgia, myalgia, a decrease in memory, loss of concentration and speech problems which last for more than 6 months. This set of complex clinical manifestations is called Post-Treatment Lyme Disease Syndrome (PTLDS). The etiology is unclear, and this syndrome is still intractable.

There are similarities between PTLDS related to LD and reactive symptoms described in BYS, and both involvements are possibly linked to immunologic disorders induced by *B. burgdorferi* infection. In Brazil, manifestations similar to those reported in PTLDS are also observed, and due to similarity in symptoms with those reported in CFS/ME, we decided to adopt the term BYS reactive stage resembling CFS/ME instead of PTLDS. It is important to note that this reactive stage is observed mainly in untreated patients.

Some BYS patients exhibit more severe manifestations, similar to those observed in autoimmune diseases, such as those displayed by patients with rheumatoid arthritis, polymyositis, localized scleroderma, encephalomyelitis, panniculitis, idiopathic vasculitis, Parkinson´s disease and amyotrophic lateral sclerosis. Other patients develop recurring skin lesions and intolerance to certain foods and medications following *B. burgdorferi* infection. This spectrum of broad manifestations in BYS patients, triggered by *B. burgdorferi* infection, is named the reactive autoimmune stage.

Based on clinical analysis and progression of hundreds of BYS patients, viewed for almost 30 years, we propose to demonstrate the disease expression and progression of this serious and complex syndrome.
Localized acute infection;Early disseminated infection;Latent infection;Reactive disease expression resembling CFS/ME;Reactive disease expression resembling autoimmune or allergic diseases;Unclassified disease expression.

#### 7.1.1. Localized Acute Infection

Nearly 40–50% of BYS patients show erythema migrans (EM) at the site of the tick bite. This typical skin lesion has an incubation period varying from 4 to 30 days, 10.8 days on average. Its duration varies from 3 days to 3 months, 27.3 days on average, and can possibly disappear spontaneously without treatment (Figure 12).

#### 7.1.2. Early Disseminated Infection

In this disease, expression of *B. burgdorferi* disseminates through the vascular or lymphatic circulation systems, and patients can present flu-like symptoms such as fever, chills, myalgia, arthralgia and headaches. Sometimes multiple and less expanding skin lesions, known as anular secondary lesions, can appear. Another type of lesion called lymphocytoma, histologically characterized by the accumulation of B lymphocytes, has also been identified in Brazilian BYS patients (Figure 13) [82,83].

Other intriguing skin lesions can be found at the site of the tick bite. On two occasions, we noticed a red expanding maculopapular eruption resembling EM, but, unlike the typical Borrelia lesion, these cutaneous injuries continue to expand for many years. The histological analysis performed in both cases revealed increased collagen deposition, suggesting localized scleroderma lesions. One of these cases showed a positive PCR test using *flgE* primer, conducted on a piece of skin margin obtained from a biopsy. It is important to specify that the patient had been treated with many courses of antibiotics (Figure 3).

During the early dissemination of *Borrelia*, articular involvement is found in 40% of BYS patients. Typically, it involves large joints, mainly the knees, with significant joint effusion (Figure 14). Synovial fluid analysis reveals the presence of an inflammatory process with an elevated number of neutrophils. Arthritis lasts a few weeks or months and disappears spontaneously without treatment. Sometimes, recurrent episodes of inflammatory articular flares develop into chronic joint involvement, similar to rheumatoid arthritis, which also involves small joints together with morning stiffness (Figure 15).

Neurological involvement is observed in nearly 35% of BYS cases. Basically, the manifestations are very similar to those observed in LD patients, which include lymphomononuclear meningitis, cranial neuritis and peripheral neuropathies. Meningitis causes slight neck stiffness and headaches, and CFS analysis reveals an increased number of mononuclear cells and hyperproteinemia, sometimes with the presence of oligoclonal bands. Since the intrathecal synthesis of antibodies to *B. burgdorferi* can be observed in BYS patients, looking for these antibodies in CSF can be helpful in diagnosing neuroborreliosis.

The pattern of sensitive and/or motor peripheral axonal neuropathy can cause intense pain, weakness or paresthesia, often of metameric distribution. Symptoms can last over time and are difficult to treat. Any cranial nerve can be involved in BYS. Bilateral facial nerve injury is highly suggestive of BYS neuropathy, as well as the simultaneous presence of meningitis in addition to the lesion of one or more cranial nerves.

Psychiatric manifestations are common in BYS patients, representing nearly 20% of all neuro-psychiatric cases. Symptoms vary from mental depression to complex and severe manifestations such as psychotic episodes, schizophrenia and suicide attempts.

Shinjo et al. (2009) [69] reported 30 cases of neuroborreliosis in BYS patients, which included 15 cases of meningitis (50%), 16 patients who presented peripheral neuropathies (53.3%), 13 who had cranial neuritis (43.3%), 10 encephalitis (33.3%), 6 reported psychiatric disorders (20%), 2 had myelopathies (6.6%), 2 had myositis (6.6%) and convulsions were reported in 1 case (3.3%). Only half of the patients remembered having been bitten by a tick or developing an EM prior to the onset of the neurological symptoms. Half of the patients with neuroborreliosis presented concurrent reactive arthritis, and 75% developed relapsing symptoms, not necessarily neurological ones.

It is interesting to notice that a large number (nearly 30%) of cases of neuroborreliosis showed symptoms suggestive of encephalitis. Some of these patients experienced severe and repeated episodes of headaches, often followed by muscular contractions or fasciculation, gait disturbances, loss of equilibrium and cognitive changes. Infectious encephalitis is generally difficult to diagnose; CFS analysis is necessary and can show an increased amount of protein, sometimes oligo clonal bands are also seen, and the identification of antibodies against *B. burgdorferi* is also possible. Magnetic resonance sometimes shows small foci of demyelination. Patients with encephalitis do not respond well to treatment and generally develop symptoms typical of chronic fatigue syndrome.

BYS patients can progress, developing severe neurological manifestations such as Guillain–Barre syndrome and unusual presentations resembling amyotrophic lateral sclerosis, multiple sclerosis or Parkinson´s disease. However, in such circumstances, we must take into account whether these kinds of neurological involvements are the consequence of an ongoing infection or they may be the result of an autoimmune disorder triggered by *B. burgdorferi* crossing the blood–brain barrier and spreading to the central nervous system. Making therapeutic decisions under such circumstances is difficult, depending on the severity of the disease, its evolution over time, previous treatments and the involvement of others organs systems. Treatment methods vary and may include antibiotics as well as intravenous gamma globulin, immunosuppressive agents or biological therapy.

We wonder if the pleomorphic atypical morphology of *B. burgdorferi* found in BYS patients could contribute to the rise and progression of severe cases of neuro-psychiatric Borreliosis observed in Brazil. It may be that these unusual spirochetes found in the central nervous system are more difficult to treat since they express fewer antigens and, thanks to their ability to survive inside host cells, they manage to evade the host’s immune system and antibiotic therapy.

Garin–Bujadoux–Bannwart syndrome, which is an association of meningitis with very painful peripheral neuropathy caused by *B. garinii*, is described in Europe but not observed in Brazil.

Cardiac involvement among BYS patients is observed in less than 5% of cases. Similarly, with LD abnormalities, the heart can show varying degrees of disorder involving atrioventricular electrical stimulus conduction. Clinical recovery does not require antibiotics administration or pacemaker implantation.

One interesting aspect observed in BYS patients is the high incidence of ocular involvement. In addition to disorders related to abnormalities of the ocular, causing diplopia, anisocoria, eyelid ptosis, squinting, mydriasis and ophthalmoplegia, we can identify intrinsic ocular lesions such as uveitis, retinal arteritis and papilledema.

#### 7.1.3. Latent Infection

Assays employed for PCR amplification and identification of *B. burgdorferi* in BYS patients can confirm the persistence of spirochetes in many tissues and organs. As evidence, we will report three cases here below. The first case is that of a woman bitten by a tick on her forearm, who presented a classical EM that improved after treatment with one month of doxycycline. However, one year later, she exhibited an identical skin lesion in the same place as the initial skin injury. She was cured after a second and longer course of antibiotics.

Another patient, who worked as a beekeeper, presented with a tick still attached to the skin on her abdomen. A few days later, a red, progressive, growing macular skin lesion developed at the site of the tick bite. Despite many antibiotic cycles, the lesion continued to enlarge, and a skin biopsy was performed at the margin of the wound four years later. A PCR test using *flgE* primer demonstrated the persistence of *B. burgdorferi* infection, and a histopathological exam showed a great deposit of skin collagen, compatible with localized scleroderma (morphea) or acrodermatitis chronica atrophicans (ACA).

The last case is that of a man who reported an EM following a tick bite in Manaus County, Amazon State, in the Amazon Forest. He was treated with doxycycline for 30 days and was declared cured. Six months later, he presented with knee arthritis and joint effusion. A PCR was performed, which showed positivity, proving that the *Borrelia* species present in the country causes relapsing episodes and needs longer periods of antibiotic treatment.

The first cases of BYS were treated according to the Center for Disease Control and Prevention guidelines that recommend less than one month of antibiotic treatment. Surprisingly, we observed that more than half of the patients who were treated following this protocol presented with subsequent relapsing episodes. For this reason, we decided to treat the BYS patients for at least three months and mostly in combination with hydroxychloroquine, a drug with immunomodulation activity. It is possible that longer courses of antibiotic therapy in BYS patients are necessary due to the presence of the atypical forms of *Borrelia* found in the country.

Cases such as these make us wonder how many BYS cases go undiagnosed and how many patients with clinical manifestations resembling connective tissue diseases, chronic neurological disorders, mental illnesses or chronic fatigue syndrome are misdiagnosed while, in fact, they are suffering from BYS. We must remember that less than 40% of BYS patients present an EM at the early stage of the disease-causing considerable diagnostic difficulty.

#### 7.1.4. Reactive Disease Expression Resembling CFS/ME

Symptoms related to chronic fatigue syndrome/myalgic encephalitis are commonly found in BYS patients, and therefore, these complaints are regarded as minor criteria for the diagnosis of Brazilian Borreliosis. In 2009, Bonoldi [70] reported the presence of CFS/ME symptoms in 38 out of 70 cases (54.3%), while in 2000, Mantovani [34] demonstrated the presence of the disease in 40 out of 68 (58.8%) patients. The Center for Disease Control and Prevention adopts the diagnostic criteria suggested by Fukuda et al. (1994) [71], which require the presence of severe fatigue for more than six months in conjunction with four of the following symptoms: post-exertional malaise; unrefreshing sleep; impaired memory or concentration; myalgia, arthralgia or a sore throat; tender lymph nodes or headaches.

In 2015, the National Academy of Medicine suggested a new diagnostic framework for CFS/ME [72].

Diagnosis requires the following three symptoms:Severe fatigue that persists for longer than 6 months;Post-exertional malaise;Unrefreshing sleep.

Diagnosis also requires at least one of the two following manifestations:Cognitive impairment;Orthostatic intolerance which can be induced by stress, infections, endocrine disorders, sleep problems and medications.

The pathogenesis of CFS/ME is unknown and can be triggered by stress, infections, endocrine disorders, sleep disorders or medications. Many infectious agents are postulated as possible microbial triggers of CFS/ME symptoms, including *Epstein–Barr* virus (EBV), *Coxiella burnetii*, COVID-19, Influenza H1N1, Chikungunya, Dengue and Ebola. Wong et al. (2021) recently suggested that infectious agents can initiate autoimmunity and could also be responsible for the onset of chronic fatigue syndrome [73].

There is significant evidence indicating that CFS/ME can be linked to abnormalities in the immune system. Wirth et al. (2021) [74] recently detected autoantibodies against β2-adrenergic receptors (β2ADr), which are responsible for many of the alterations observed in CFS/ME which are related to autonomic nervous system dysfunction, sympathetic over activity that results in excessive vasoconstriction affecting the brain and skeletal muscles. In contrast, the generation of vasodilators causes hypovolemia and renin suppression.

#### 7.1.5. Reactive Disease Expression Resembling Autoimmune or Allergic Diseases

Over time some BYS patients reveal signs of autoimmune or allergic diseases, presenting clinical manifestations which resemble Sjögren’s syndrome; rheumatoid arthritis; inflammatory myositis similar to poly-dermatomyositis with increased muscle enzymes (creatine phosphokinase, aldolase) and electromyography alterations; panniculitis; localized scleroderma; venous thrombosis with antiphospholipid antibodies; clinical manifestations mimicking systemic lupus erythematosus such as Raynaud’s phenomenon, alopecia and photosensitivity; sudden onset of drug or food allergies.

Laboratory abnormalities are also observed in some BYS patients following *B. burgdorferi* infection, for example, hypergammaglobulinemia; antinuclear antibodies (ANA); anti-Ro/La; anticardiolipin IgG/IgM; antineutrophil cytoplasmic antibodies (ANCAs); elevated IgE immunoglobulin.

A very common and peculiar finding observed in BYS is the appearance of relapsing skin lesions, resembling EM, or multiple and disseminated maculo-erytematous lesions, which could be confused with the symptoms of the early stage of Borrelia *burgdorferi* infection (Figure 16 and Figure 17). These reactive skin lesions are not generally followed by flu-like symptoms or systemic manifestations. In most cases, they appear when patients are debilitated, or can be triggered by physical stress such as exposure to high temperatures, for example, after having a hot bath. These types of skin lesions have not been reported in LD patients so far; therefore, we decided to describe them as reactive exanthema.

The etiology of reactive exanthema and other immunological abnormalities observed in BYS is unknown. According to Eskandari et al. (2003) [75], there is a strong connection between the immune system and the central nervous system (CNS) through the release of immune-regulatory hormones such as corticotrophin, cortisol and vasopressin, which regulate the hypothalamus–pituitary–adrenal axis. *B. burgdorferi* can cross the blood–brain barrier, activating microglia cells which cause the release of inflammatory cytokines (IL-6, IL-8, IL-12, IL-18 and IFN-ɣ) and chemokines (CXCL-2 and CXCL-13), which in turn, activate the hypothalamus–pituitary–adrenal axis. Imbalances of this modulatory system could predispose the onset of inflammatory, autoimmune, infectious or allergic diseases.

Inflammatory cytokines and chemokines alone can also be responsible for symptoms of CFS/ME, so testing these inflammatory mediators, especially in the cerebrospinal fluid, can be a useful parameter to quantify disease activity in the central nervous system.

Arvikar et al. (2017) [76] were the first LD research group to suggest the onset of systemic autoimmune rheumatic diseases, such as rheumatoid arthritis, psoriatic arthritis or peripheral spondyloarthropathy, triggered by Lyme disease. The authors discuss whether this finding is a mere coincidence since these autoimmune diseases are relatively common, and LD is endemic in the region. Another possibility taken into account by the authors is that *Borrelia* infection might convert the preclinical phase of autoimmune diseases into an active rheumatic illness.

In an attempt to understand the occurrence of autoimmune features in BYS, we attempted to identify a laboratory marker that could tell us which clinical manifestations the antibodies are related to and at which stage of the disease they occur.

We examined the frozen sera of 98 BYS patients, most of whom were female (n = 72; 73.4%) with a mean age of 35 and who had been monitored for about 2.5 years on average. Clinically, 58 cases presented an EM rash (59.1%), 62 showed symptoms of arthritis (63.2%) and 67 presented symptoms indicating neurological involvement (68.3%). The sera samples of these patients were arbitrarily divided into two groups: the early stage group, formed of 20 samples (20.4%) deriving from patients who experienced a tick bite less than three months earlier, and the late stage group, comprising of 78 sera samples (79.6%) from patients who had been infected for more than three months. The control group consisted of 40 sera samples obtained from healthy subjects. ELISA assays were performed to detect IgG antibodies against two human antigens: the brain cell membrane proteins (BCMPE) (Sigma, St Louis, MI, USA) and skin tissue protein extract (STPE) (Sigma, St Louis, MI, USA). Fifty-eight sera samples (59.2%) showed antibodies against BCMPE and thirty-six (36.7%) against STPET. No individual belonging to the control group tested positive (Figure 18).

Both groups of BYS patients presented antibodies against the two human antigens. The sera of patients belonging to the latency stage group showed higher statistically significant levels of autoantibodies against brain cell membrane proteins when compared to the sera of patients belonging to the early stage group. The test also revealed the presence of autoantibodies among patients who had distinct clinical manifestations, including patients with cutaneous, articular and neurological symptoms. Patients that presented EM rash showed slightly lower levels of autoantibodies, possibly because this group included cases of isolated EM and cases of EM in association with systemic symptoms (Table 4).

#### 7.1.6. Unclassified Disease Expression

Since the HCFMUSP was the Reference Center for Investigation of Brazilian Borreliosis, a great number of patients coming from all Brazilian States were referred to us for follow-up care. For this reason, we encountered unusual and challenging cases of BYS, which were often impossible to classify in terms of disease stage. Consequently, it was just as hard to predict disease progression and the most suitable treatment for the patients. Most of these patients displayed clinical manifestations which evolved into severe and unremitting neuro-psychiatric disorders.

Due to its relevance, we decided to briefly describe the clinical history and evolution of six patients who displayed severe mental and neurological complications related to *B. burgdorferi* infection in Brazil. Long-term monitoring of these patients, which sometimes takes place over the course of years, certainly can help in understanding how difficult it can be to determine the disease stage as well as linking it to the initial *B. burgdorferi* infection. For these reasons, we decided to present only six BYS patients with a definite diagnosis and who, at the early stage of the infection, presented with EM at the site of the tick bite.

Case #1: 15-year-old girl. While hiking, the girl was bitten by multiple ticks in Colatina County, Espirito Santo State. A few days later, she presented an EM at the site of one of the tick bites. She experienced vomiting, headaches, fever, neck pain and arthralgia. The Elisa serology for *B. burgdorferi* was negative, but two IgM and one IgG bands were present, which is considered a positive result according to the HCFMUSP criteria. She was treated with doxycycline for 30 days. After 5 months, she continued to experience fevers, lumbar pain, weight loss, headaches, cervical pain, fatigue, dizziness, diarrhea, fainting spells and severe depression. The patient had to interrupt her studies due to the illness. Despite this, the ELISA test was still negative, and the WB exhibited no IgM bands but five IgG bands. The patient was treated with one month of ceftriaxone. She improved, but after three months, she experienced a relapse of severe depression, which led to a suicide attempt. Unfortunately, even after years, her symptoms have not improved; she is still affected by severe depression and requires psychiatric assistance. She has not returned to school and keeps to her room.

Case #2: 35-year-old man. The man was bitten by a tick in 2006 in São Luiz County, Maranhão State. Two years later, he presented with an expanding skin lesion that reached 15 cm in diameter at the original site of the tick bite (this is the second report of a patient with an EM at the site of the tick bite after a long latency period). The EM-like lesion lasted for around 15–20 days and was accompanied by fever, fatigue and generalized pain. Concurrently, the patient began having difficulty moving his left leg; he also experienced tingling in the fingers of his right hand and tremors in his legs and left hand. He also reported stiffness, weakness and limited movement in his left shoulder, severe dorsalgia, significant fatigue, sore throat, diffuse myalgia and polyarthritis. The patient underwent serological testing for *B. burgdorferi* infection at the HCFMUSP. The ELISA was positive (IgM negative; IgG 1/1600), and the WB showed two IgM and two IgG bands. Cerebrospinal fluid analysis was normal, the brain MRI showed no abnormalities and the electromyography showed motor unit potentials. Following the diagnosis of BYS, the patient was treated with ceftriaxone 2 g/day for 30 days but did not experience any improvement. He was prescribed additional rounds of antibiotics in combination with hydroxychloroquine. Again, there was no sign of improvement. Currently, his condition continues to worsen, showing progressive and diffuse muscle spasticity and atrophy, uncoordinated movements, difficulty swallowing and tremors. Medications to control Parkinson´s disease have been prescribed by his neurologists.

Case #3. 34-year-old woman, veterinarian. In 2002, after removing several ticks from a stray dog in Santa Maria County, Rio Grande do Sul State, the woman reported an EM on her left forearm. A few days later, she developed symptoms of an auditory disorder and weakness. After another two months, the patient showed psychiatric symptoms of schizophrenia and bipolar disorder, which were treated with antipsychotic medications. From this moment onwards, she also presented recurring cutaneous skin lesions and polyarthritis. The rheumatologist then diagnosed her with BYS and prescribed ceftriaxone in combination with penicillin which led to a partial improvement of symptoms. However, the psychiatric symptoms persisted. In 2009, following a yellow fever vaccination, the patient experienced a worsening of her mental state and also presented a flare-up of arthritis in her hand. She improved on doxycycline. In 2010, the patient experienced another episode of intensified arthritis, which again improved with antibiotic treatment. In October 2011, she complained of a further relapse with severe headaches, paresthesia of the feet, light sensitivity, blurred vision, taste alteration and severe fatigue. Brain magnetic resonance imaging (MRI) showed normal results.

Case # 4. 15-year-old girl. The patient was bitten by a tick in Bonito County, Mato Grosso do Sul State. She developed an EM and was treated with ceftriaxone in combination with hydroxychloroquine. A few months later, she started suffering from depression and temper tantrums and began deliberately hurting her own body by cutting herself with sharp objects. Additionally, she reported episodes of low fever, polyarthritis, headaches, fatigue and numbness in her feet. She was admitted to the hospital several times, and during one of these hospitalizations, she attempted suicide. Five years have passed since she was infected with Borreliosis, and she is still experiencing flare-ups of psychiatric symptoms.

Case # 5. 13-year-old girl. In March 2012, the patient visited a small farm that kept horses. A few days later, she developed an EM on her abdomen, followed by other skin lesions on her body. At the same time, other symptoms began appearing, such as severe headaches, fever, neck pain, diplopia, loss of balance, paresthesia in hands and feet, weakness and muscle tremors. Because her symptoms continued to progress during following two months, she was hospitalized for 32 days and was diagnosed with BYS. Her serological tests for *B. burgdorferi* were positive, and CSF analysis showed elevated numbers of mononuclear cells, which confirmed a diagnosis of meningitis. Brain Magnetic Resonance Imaging showed normal results. She was treated with ceftriaxone for 30 days, followed by 21 days of doxycycline. Despite the antibiotic treatment, the patient complained for months of headaches, loss of strength in her arms and legs, weakness, irritability, sleeplessness, tinnitus and presented arthritis of the knees, feet and elbows. Due to the severity of the illness and the lack of response to further antibiotic cycles, she was prescribed pulse therapy with methylprednisolone, which also brought little improvement. Since the patient first fell ill, she was frequently hospitalized. In an attempt to control the flare-ups of neuro-psychiatric symptoms, lacking other therapeutic options, she was repeatedly treated with antibiotics, analgesics and anti-depressants. Since there was no improvement, intravenous gamma globulin and immunosuppressive agents, such as azathioprine, were administered in an attempt to improve her health condition. Unfortunately, she did not recover. Despite being treated and monitored over the course of her illness she is still experiencing flare-ups of unremitting headaches, double vision, intense fatigue, neck pain, cognitive disturbances, sleep disorders, fainting spells and light and sound sensitivity.

Case # 6. 56-year-old woman. The patient recalled being bitten by a tick on her abdomen in 1986 at Serra do Araripe, Crato County, Ceará State, and then developing an enlarging skin lesion at the site of the bite, accompanied by fever and arthritis. After about two years, since the onset of her first symptoms, she started to experience speech difficulties, loss of balance, muscle spasticity, stiffness of the fingers and recurrent erythematous skin lesions. The various neurologists she consulted diagnosed her with atypical Parkinsonism and amyotrophic lateral sclerosis. In spite of the many medications that were prescribed, the neurological symptoms continued to progress. Coincidently, the patient’s daughter had a recollection of the skin lesion reported by her mother, which she associated with Lyme disease and assumed that her mother might have Borreliosis. The patient´s clinical signs corresponded to BYS. Although the ELISA test was negative, the WB, according to the HCFMUSP criteria, was positive, showing two IgM and four IgG bands. Brain Magnetic Resonance Imaging showed normal results. She was treated with ceftriaxone for 30 days, followed by two months of doxycycline in combination with hydroxychloroquine. The neurological symptoms did not disappear; however, disease progression ceased.

From the analysis of these six cases of unremitting neuropsychiatric manifestations following *B. burgdorferi* infection, we came to the following conclusions:

*B. burgdorferi*, the etiological agent of BYS, can remain dormant for long periods of time (cases # 2 and 6) and progress into the tertiary stage of the disease. This stage can be defined by a wide range of symptoms that develop after months or years after the initial Borrelia infection in a person who has not been treated with antibiotics. Diagnosis of late stage neuroborreliosis generally leads to little improvement even after antibiotic treatment since spirochetes may cause chronic structural derangements in the brain with irreversible damage to the central nervous system. Chronic encephalitis may be a consequence of infection caused by *Borrelia burgdorferi*; however, inflammation and injury in the brain may be a result of autoantibodies [77].

On the contrary, patients one, three, four, and five developed fast and progressive damage to the central nervous system soon after the acute stage of *B. burgdorferi* infection. Coincidentally, all four female patients, two of them young students, presented severe and unremitting psychiatric involvement, which did not respond to antibiotic treatment. The clinical analysis of each case suggests that the patients suffered a form of acute and severe encephalitis, which became progressively chronic and irreversible. Infection-induced autoimmunity, triggered by *B. burgdorferi*, was also taken into account as a possible cofactor which may lead to worsening and persistent brain damage. Furthermore, many of the severe and ongoing disabilities reported by patient five could be related to symptoms of chronic fatigue syndrome, which are also a consequence of *Borrelia* infection, according to our research.

The first studies on autoimmune encephalitis were conducted by Dalmau et al. (2007) [78], who described the autoantibodies which affect the antigens responsible for synaptic transmission localized in the cell membrane. Later, Jammoul et al. (2016) [79] observed that different clinical presentations of autoimmune encephalitis depend on the brain region that is affected by the autoimmunity. Therefore, limbic encephalitis is mainly characterized by short memory loss, agitation, confusion, depression, psychosis and relapsing convulsions. Involvement of the brainstem (mesencephalon and medullar bridge) usually develops into changes in ocular movement, eyelid ptosis, dysphagia, dysarthria, ataxia, facial palsy, dizziness, hearing loss, decreased consciousness and hyperventilation. Encephalitis involving the cerebellum in general has long-term clinical consequences, such as dizziness, limping and ataxia.

Despite the different etiopathogenesis of the encephalitis observed in classical autoimmune conditions and that develops post-infection [77], there are strong similarities in the clinical presentation of both kinds of encephalitis. This suggests that the range of involvement of the central nervous system observed in BYS can vary depending on the region where the lesions caused by *B. Burgdorferi* are located and also supports a possible role for autoimmunity in the mechanisms of neurological involvement by *Borrelia* infection.

## 8. Treatment of Baggio–Yoshinari Syndrome

In general, BYS is treated with the same medications used for LD. For the early localized stage (EM alone), treatment consists of the use of doxycycline 100 mg orally twice a day, amoxicillin 500 mg 8/8 h, azithromycin 500 mg once a day or cefuroxime 500 mg twice a day for 30 days.

For the disseminated infection stage, therapy involves the use of the same drugs for an extended period of three months. In the presence of neurological symptoms, it is preferable to use ceftriaxone 2 g administered intravenously for 30 days, followed by two months of additional oral therapy with doxycycline. It is possible to substitute ceftriaxone with crystalline penicillin, with 4 million units administered every 4 h.

In most cases, physicians from FMUSP are accustomed to combining the above treatment with oral hydroxychloroquine 400 mg prescribed for a period of 6 months. It was believed that if this drug was prescribed during the disseminated stage of infection, it could modulate the immune-mediated inflammatory response, blocking the release of pro-inflammatory cytokines such as IL-6.

Patients with BYS in the reactive stage presenting symptoms of CFS/ME are difficult to treat. They do not respond well to medications such as analgesics, anti-inflammatories and anti-depressants, and sleeping pills are prescribed only to alleviate the discomfort associated with the many non-specific symptoms. In general, since these medications do very little to further the improvement of the patients, supportive therapies such as physical therapy, massage, acupuncture and myofascial release are also recommended.

Treatment of the reactive stages resembling chronic rheumatic diseases, such as rheumatoid arthritis, can require the use of disease-modifying anti-rheumatic drugs e.g., hydroxychloroquine, methotrexate, sulfasalazine and sometimes corticosteroids or biological medications.

For patients with suspected encephalitis which does not respond to antibiotics, treatment should involve drugs to help ease their symptoms, while, if symptoms should worsen, it is possible to administer gamma globulin or immunosuppressive agents intravenously.

## Figures and Tables

**Figure 1 pathogens-11-00889-f001:**
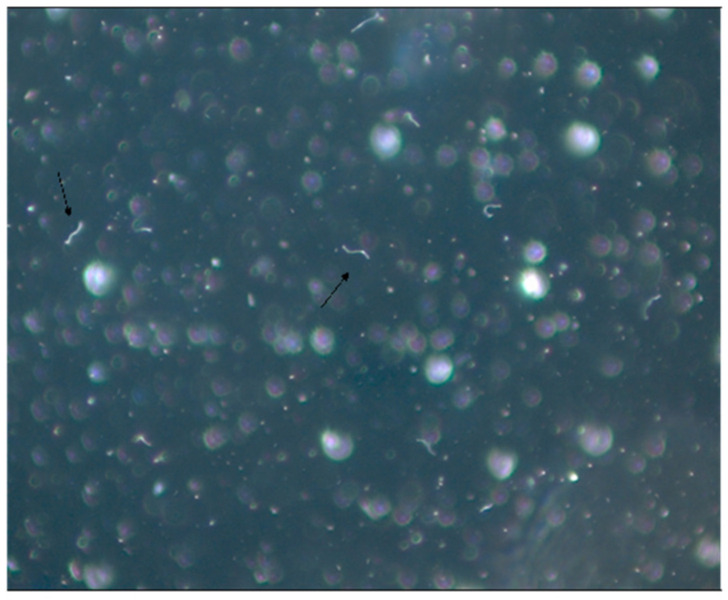
Patient’s blood culture in SP4* medium viewed on dark field microscopy (1000×). Mantovani et al. (2007). * SP-4 medium is used for Spiroplasma culture.

**Figure 2 pathogens-11-00889-f002:**
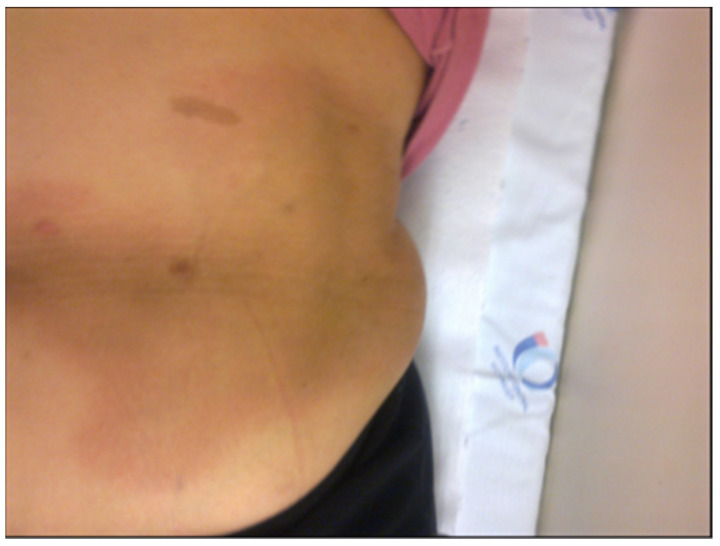
Persistent and expanding skin lesion in BYS patient, lasting more than four years at initial tick bite site. PCR test positive with *flgE* primer even after many antibiotic shots.

**Figure 3 pathogens-11-00889-f003:**
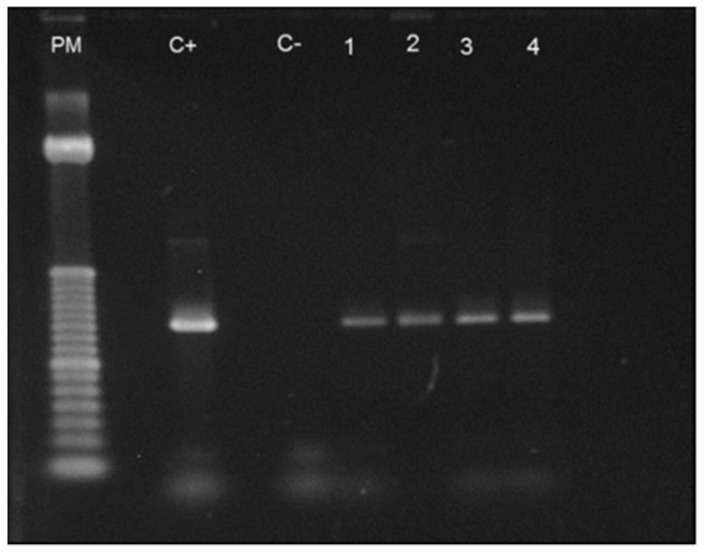
PCR performed with flgE primer to identify positive human BYS cases. PM = molecular weight (50 bp); C+ = *B. garinii**;* C− = negative normal control; 1 to 4 = patients with BYS.

**Figure 4 pathogens-11-00889-f004:**
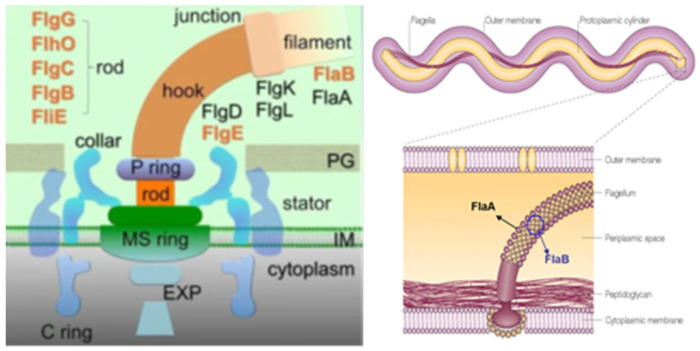
Flagella apparatus of *B. burgdorferi*: basal body, hook and flagella filament. PCR test in Brazil was conducted using primer derived from flagella hook (flgE) [42,43].

**Figure 5 pathogens-11-00889-f005:**
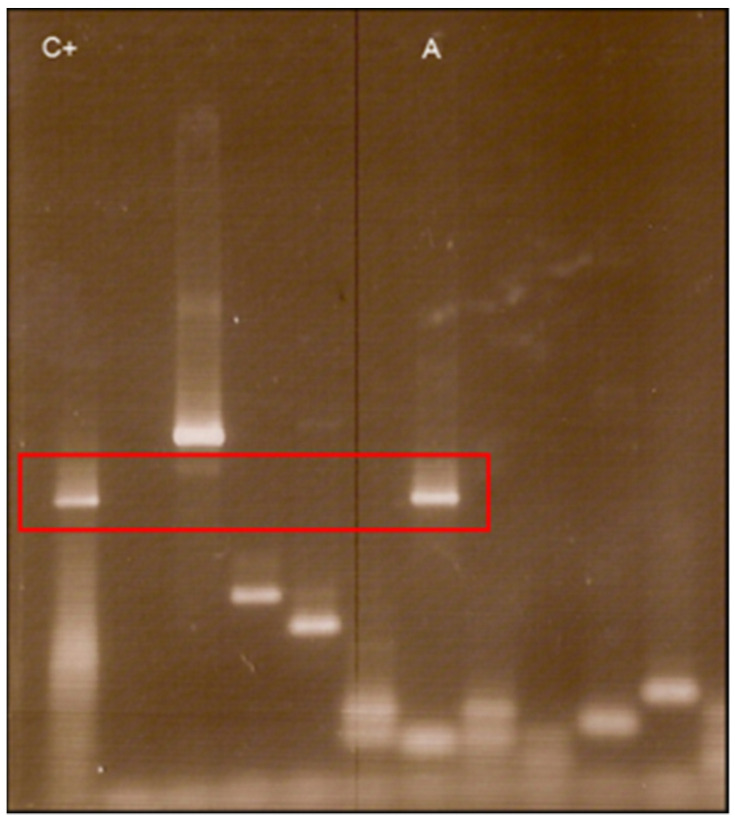
Positive PCR performed on tick DNA samples using flgE primer. C+ = *B. garinii*; A = tick sample.

**Figure 6 pathogens-11-00889-f006:**
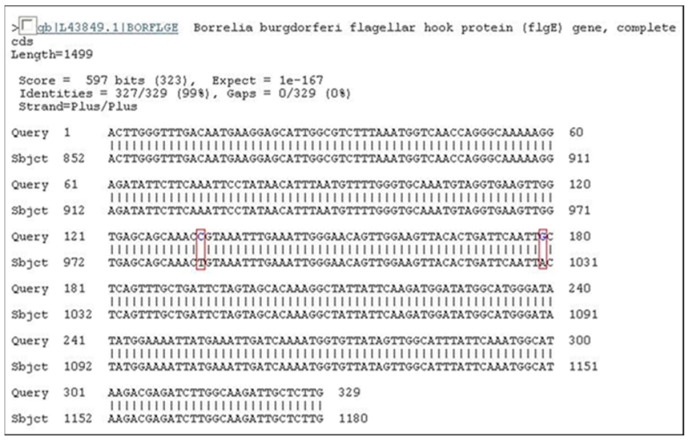
Sequencing of positive samples obtained after PCR performed with genes of *Borrelia burgdorferi* flagellar hook protein (*flgE*) (L43849) carried out in BLAST. See differences between two pairs of nucleotides in comparison with *B. burgdorferi* ss.

**Figure 7 pathogens-11-00889-f007:**
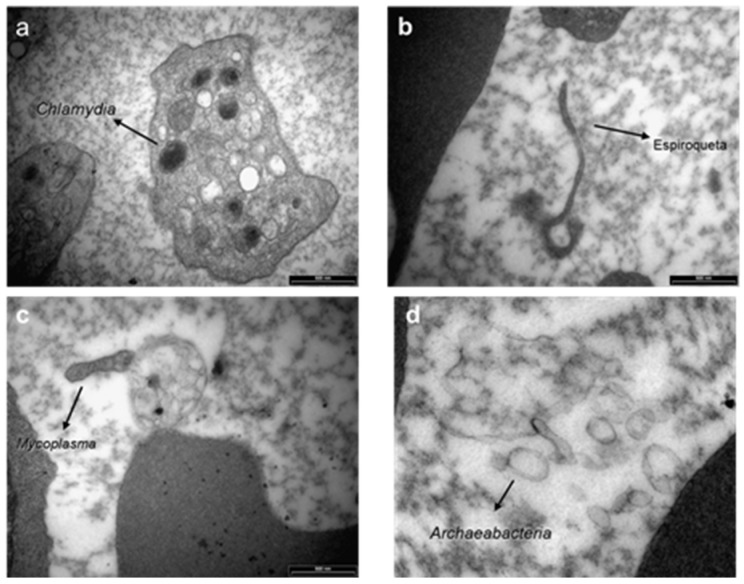
Structures resembling *Mycoplasma* spp., *Chlamydia* spp., spirochete-like and archaebacteria, viewed by electron microscopy analysis, obtained from peripheral blood of BYS patient. (**a**,**b**) 24,000× C; (**c**,**d**) 30,000×. Prof. Maria de Lourdes Higuchi (INCOR HCFMUSP).

**Figure 8 pathogens-11-00889-f008:**
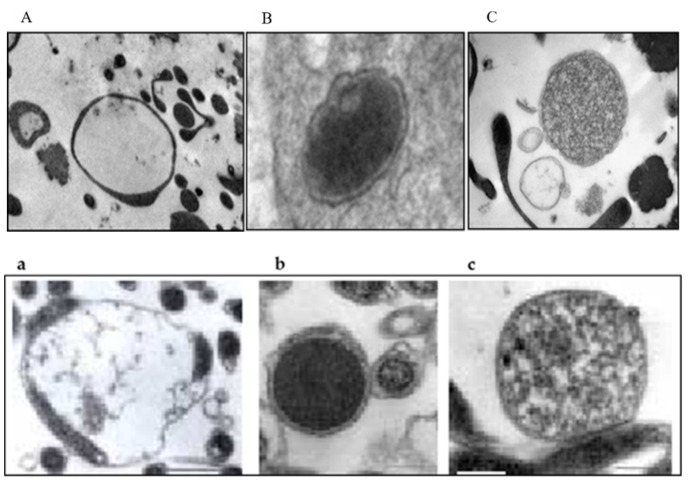
Comparison of electron microscopy analysis of morphologically atypical structures viewed in BYS patient´s blood seeded in SP-4 medium (above) and *B. burgdorferi* cultured at inhospitable conditions with antibiotics (below). (**A**–**C**): BYS patient blood sample; (**a**–**c**): Literature: Kersten et al., 1995 [52]. Vesicle-like structure (cell wall deficient bacteria) (**a**); round dense body-Chlamydia-like (**b**); Mycoplasma-like and spirochete (**c**). Prof. Maria de Lourdes Higuchi.

**Figure 9 pathogens-11-00889-f009:**
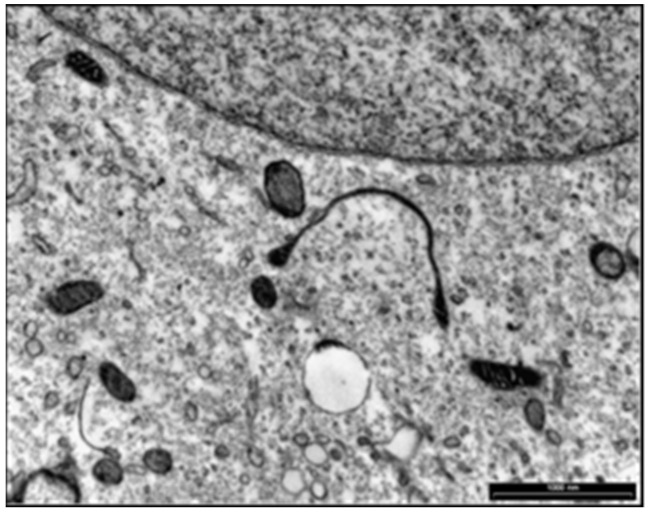
Spirochete-like structure found inside endothelial cells after inoculation of BYS patient´s blood seeded in SP-4 culture medium (15,000×). Prof Maria de Lourdes Higuchi.

**Figure 10 pathogens-11-00889-f010:**
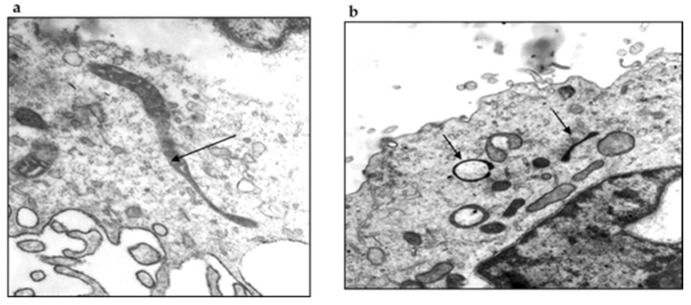
Suspension of *B. burgdorferi* G39/40 was inoculated in endothelial cell culture. After 7 days, we see modified spirochete-like structures inside cells: *B. burgdorferi* that lost its usual spiraled shape inside endothelial cells (7200×) (**a**); *B. burgdorferi* that shows spirochete-like and round cystic structures forms (4.200×) (**b**). Prof Maria de Lourdes Higuchi (INCOR HCFMUSP).

**Figure 11 pathogens-11-00889-f011:**
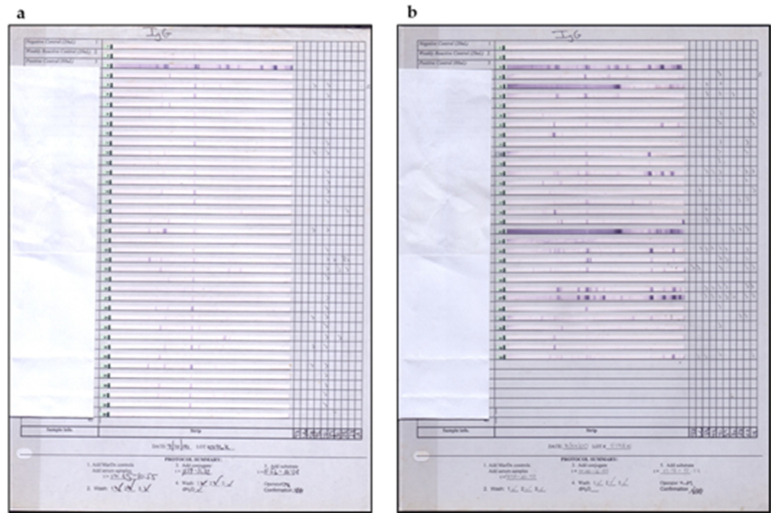
IgG WB of BYS patients (n = 36) (**a**) and IgG WB of LD patients (n = 33) (**b**). The WB assays were performed at Dr Steere’s laboratory in 2000.

**Figure 12 pathogens-11-00889-f012:**
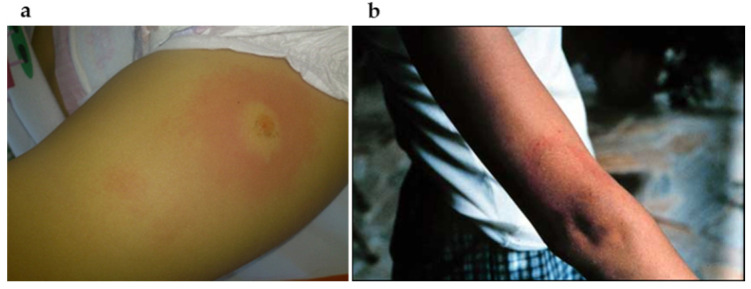
Erythema migrans lesions on arms of BYS patients.

**Figure 13 pathogens-11-00889-f013:**
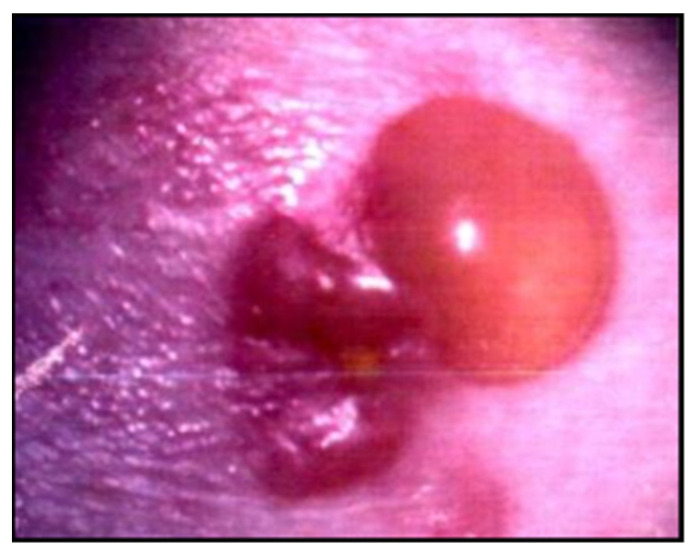
Borrelial lymphocytoma in BYS patient.

**Figure 14 pathogens-11-00889-f014:**
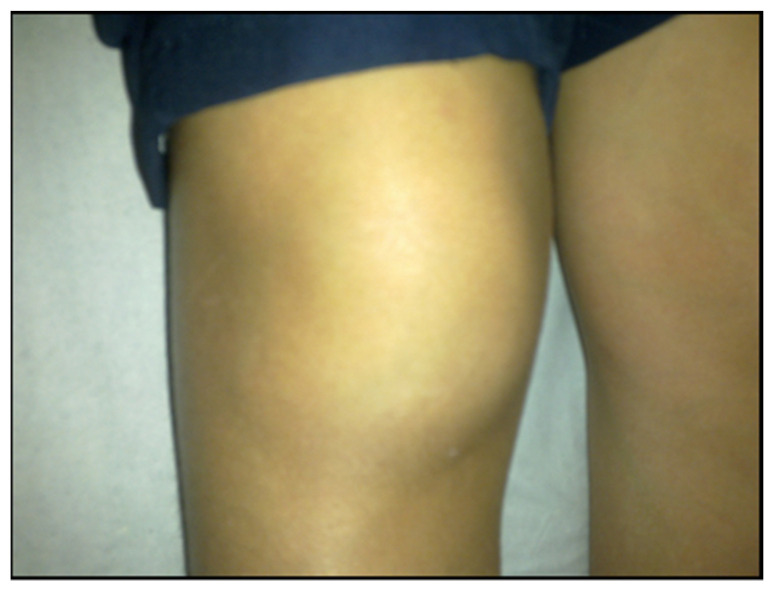
Monoarthritis with important joint effusion in child with BYS.

**Figure 15 pathogens-11-00889-f015:**
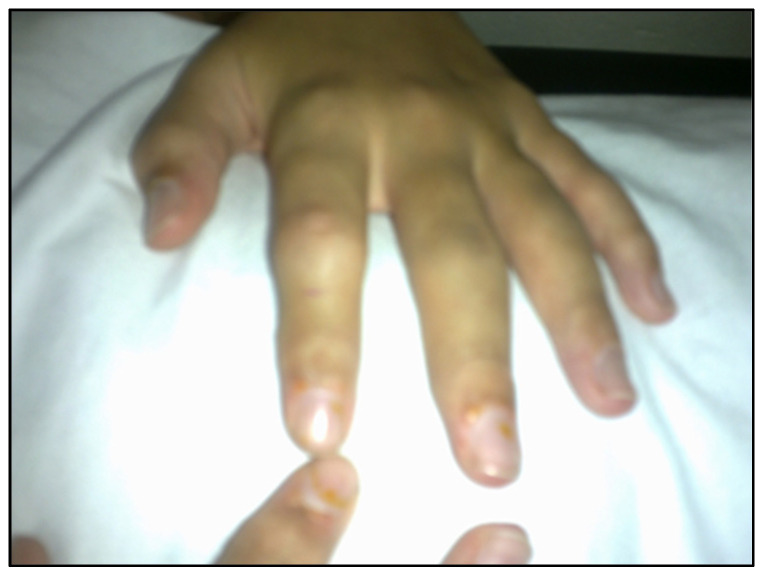
Polyarticular involvement such as rheumatoid arthritis in BYS patient.

**Figure 16 pathogens-11-00889-f016:**
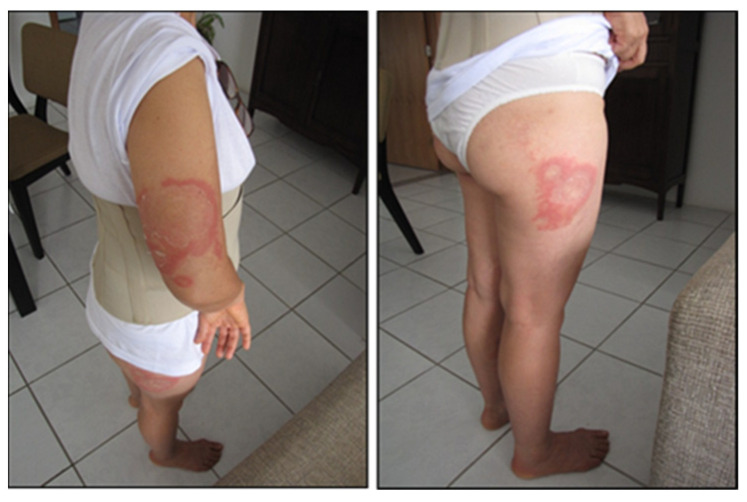
Multiple skin lesions named reactive exanthem found in a BYS patient.

**Figure 17 pathogens-11-00889-f017:**
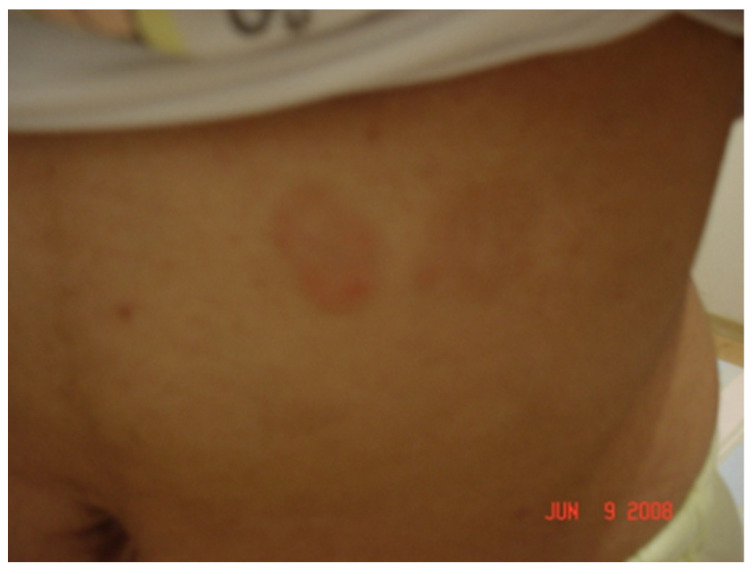
Reactive exanthem found in patient with BYS.

**Figure 18 pathogens-11-00889-f018:**
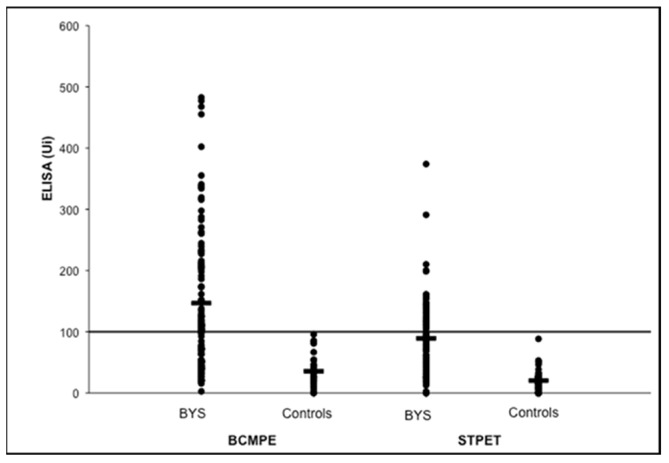
IgG antibodies to human self-antigens from sera of 98 BYS patients. BCMPE = brain cell membrane proteins; STPET = skin tissue protein extract; ELISA = enzyme-linked immunosorbent assay.

**Table 1 pathogens-11-00889-t001:** Frequencies of antibodies to *B.burgdorferi* in BYS patients according to criteria adopted at HCFMSP.

	SBY (n = 68)	Normal (n = 50)
ELISA	23 (33.8%) *	6 (12%)
Western Blotting (WB)	40 (58.8%) *	6 (12%)
ELISA and WB	19 (27.9%)	4 (8%)
ELISA plus WB	44 (64.7%) *	8 (16%)

* Statistically significant.

**Table 2 pathogens-11-00889-t002:** The frequencies of PCR positivity performed on DNA isolated from ticks, buffalos and horses using flgE primer.

	Positives (n)	Frequencies (%)
Ticks (n = 47) *	2	4.25
Buffalos (n = 27)	1	3.70
Horses (n = 26)	1	3.85

* Tick species: Rhipicephalus sanguineus and Rhipicephalus microplus.

**Table 3 pathogens-11-00889-t003:** Lyme Borreliosis Sub-Groups.

Groups	Sub-Groups	Humans	Clinical Aspect		Hosts Reservoirs	Vectors/Ticks
			Erythema migrans	Febbre		*Hard-Ticks*
Lyme Group	Organotropism	Yes	Yes	No	Rodents	*Ixodes sp.*
	High Spirochaetemia	Yes	Yes	Yes	Rodents	*Ixodes sp.*
	Baggio–Yoshinari	Yes	Yes	Yes (78%)		*Amblyomma sp.*, *Rhipicephalus sp.*

**Table 4 pathogens-11-00889-t004:** Frequencies of autoantibodies to BCMPE (brain cell membrane proteins) and STPET (skin tissue protein extract) in sera of BYS patients according to disease stage.

Stage	BCMPE	STPET
Early (n–20)	8 (40.0%)	6 (30.0%)
Late (n–78)	50 (64.1%) *	30 (38.4%)
Total (n = 98)	58 (59.2%)	36 (36.7%)

Early stage: ≤3 months; Late stage: >3 months. * Statistically significant.

## Data Availability

Not applicable.

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
