# Peer review of "The Current State of Knowledge on Baggio–Yoshinari Syndrome (Brazilian Lyme Disease-like Illness): Chronological Presentation of Historical and Scientific Events Observed over the Last 30 Years"

_pathogens, 2022, doi:10.3390/pathogens11080889_

Round 1

Reviewer 1 Report

1.      Line 46 “In this respect, since ticks of the Ixodes ricinus complex had been identified in Brazil [2].

Minor, sentence confusing.

2.      This second fieldwork conducted in Campo Grande county, state of Mato Grosso do Sul, confirmed the absence of ticks of the Ixodes ricinus complex which are usually involved as vectors in the transmission of LD in the Northern Hemisphere.

Significant need for clarification. I thought Ixodes scapularis was the main tick in the Northern Hemisphere.

3.     Line  219, “Analysis of the first Brazilian LD-like cases, showed low humoral and cellular imune response to B. burgdorferi when compared with north-american patients [15-20].”

I work in the North America. I understand the cellular and humoral response is low in the Northern Hemisphere if all patients are included. The published paper typically exclude the patients with negative tests and therefore inflate the number with humoral and cellular immune response.  The IgG western blots are often false negative.       

4.     Line 219  Imune should be immune

5.      262-267 Lyme disease initially identified as B. burgdorferi sensu strictus from the Ixodes scapularis tick. The research community was able to incorporate differing genospecies and ricinus ticks in Europe.  There were many papers describing the differences and similarities of the European findings. I feel the medical community should be inclusive of your finding rather than splintering off to Lyme-like organism.  Lyme itself is an umbrella term. The confusion is apparent with the  plethora of terms “Brazilian Lyme disease, Brazilian Lyme diseaselike illness, Lyme disease-simile illness; Infectious-reactive Lyme disease-like syndrome; and finally Baggio-Yoshinari Syndrome (BYS) .”

6.     279 publication instead of publication

7.     519   The following statement supports the conclusion that difference in ticks and infections are due to changes over time in different parts of the worlds. “In our opinion, these camouflaged Borrelia burgdorferi sensu stricto, exhibiting pleomorphic atypical morphologies, possibly appeared in our country, as an adaptative process to survive the conditions of Brazil’s ecosystem.”

8.     526 The following statements add support the Lyme umbrella “According to Meriläinen et al, 2015 [53], pleomorphic morphological forms of B.burgdorferi sensu lato, induced by different culture conditions, cause the presence of the spherical round bodies, that keep intact their flexible cell envelope. These structures should be considered clinically relevant, deserving novel diagnostics methods and treatment protocols.” And “Rudenko et al, 2019 [54], comment that survival of spirochetes from B.burgdorferi sensu lato complex in a hostile environment is achieved by the regulation of different gene. expression, in response to changes in temperature, nutrients, hosts or vectors. These altered conditions cause changes in gene expression, giving origin to persisters or dormant cells subpopulations which require low metabolic ativityies, permiting long periods without replications.”

9.     Line 597 The following attempt at subgroups highlights the public health nightmare and difficulties for doctors understanding the criteria  “6. Borreliae Lyme Group  The Borrelie Lyme Group: we can distinguish the following subgroups (Table 3) [58]: Classical Organotropic form, with EM and transmitted by Ixodes sp. Spirochaetemica Group (Borrelia mayonii) with EM transmitted by Ixodes sp. Baggio Yoshinari form with Headache and major immune reaction and transmitted by Amblyomma cajennense.”  The authors added “Clinical aspects of BYS are very similarto those presented by LD in the USA and Europe, despite the many epidemiological and laboratorial diferences observed between the two diseases: “

10.  Line 617 autoimmune and chronic fatigue syndrome symptoms are common in the Northern Hemisphere. Please clarify. “Currently, Baggio-Yoshinari syndrome is defined as an emerging tick-borne disease, up until now, specific to the Brazilian territory, caused by B. burgdorferi sensu stricto, found in its pleomorphic morphologies, non isolated nor cultivated yet, transmitted by hard Ixodid ticks not belonging to the Ixodes ricinus complex, which reproduces a clinical picture that resembles Lyme disease, except for its tendency to present autoimmune and chronic fatigue syndrome symptoms.

11.  Line 625  The following rates of positive tests are higher than the Northern Hemisphere. “In spite of this, ELISA and WB tests performed as recommended by HCFMUSP can be helpful to identify around 65% of BYS suspected patients”

12.  Line 693 I am concerned with the utility of stages. The Northern Hemisphere doctor initially tried to come up with stages but too many patients had both neurologic and rheumatologic presestations.

1- Localised acute infection stage.

2- Early disseminated infectious stage.

3- Latent infectious stage.

4- Reactive stage resembling CFS/ME.

5- Reactive stage resembling autoimmune or allergic diseases.

6- Unclassified disease stage.

13.  I might suggest a limitation section that would address the concerns in these comments. Or I would drop the term Baggio-Yoshinari Syndrome (Brazilian Lyme Disease- like illness).

Author Response

Dear reviewer of the manuscript The Current State of Knowledge on Baggio-Yoshinari Syndrome (Brazilian Lyme disese -like illness ). Chonological presentation of historical and scientific events observed over the last 30 years. 

Thanks a lot for your valuable comments and suggestions. We made the proposed changes to become the manuscript clearer and to facilitate the comprehension of the manuscript. Corrections and changes were done in text body in red (the manuscript is attached). 

We prefer not to drop the term Baggio-Yoshinari Syndrome , because currently, this nomenclature is well accepted by Brazilian physicians. Furthermore, the use of this eponym, allow to make better distinction between Lyme disease and Brazilian exotic Borreliosis. 

Sincerely, 

The authors

Reviewer 2 Report

This reviewer sees 3 possible options for how this work may be accepted for publication.  The choice amongst the three options, which the reviewer will 'lay out' is up to the authors and the Editors to decide jointly.

Option 1:  English language & syntax corrections to more proper usage, but otherwise without any major changes to the 'line of reasoning' and major 'thrust' of the authors in the mss. 'as is'.

Option 2: As above, but with more complete discussion of whether or not a diagnosis of Borrelia burgdorferi sensu stricto can presently be supported given the evidence thus far developed and the 'gaps' in the evidence (and the possibilities of with further scientific study over time) filling in those gaps to be sure that the conclusions reached are justifiable.  This will be elaborated on in the detailed review of the mss. on a line by line basis

Option 3: major revision of the text to incorporate more recent additions to the medical and scientific literature and to synthesize that in the the original work of the authors in a way that would be (to this reviewers way of thinking) add merit to the mss., make it more expansive in scope and less restricted by conventionally accepted beliefs (or what could be considered dogma).

Line by line comments, corrections and  observations follow:

Abstract:

Line 14: "...from the Northern Hemisphere..." (insert the)

Line 14:  "..that belong to the Borrelia..." (insert that and to the)

Line 16: "....symptoms characterize LD clinically." (delete the before LD and add ally, clinically)

Line18: "....findings research into BYS has been developing...."

Line 19:  "...years and show that....laboratory features...."

Line 23:  "...defined as an exotic and...."

Line 25:  "...spirochetes belonging to the B. burgdorferi sensu lato...."

Line 32: "...clinical, laboratory and therapeutic aspects..."

Line 39:  "...of this astonishing illness...."

Line 47:  "...identified in Brazil [2], Dr. Steere...." (change period to a comma)

Line 51:  "...Ixodes scapularis...."

Line 53: "....North American...."   (capital N and A)

Line 80: "...enormous data base..." (suggested instead of the term casuistic)

Line 96:  "...doxycycline...."

Line 112:  "....sex determined and..." (rather than 'sexed')

Line 112:  "...euthanized...." (suggested rather than 'put down')

Line 116:  "....animal blood...." ( or perhaps "animals' blood"?)

Line 129:  "...and do not bite humans..." (how absolute a statement is that? NEVER bite humans, or typically feed on other hosts - just wonder if that is too 'absolute' a statement and whether, on occasion, ticks of this sort might bite a human?)

Line 131: "....Amblyomma...."

Line 141:  "...exhibited positive re-.."? clarify what 'positive results' means

Line 46-147: "...these hard Ixodid ticks do not bite humans, and cannot transmit the disease." (Once again, how 'absolute' is one about this statement?  Are there 'exceptions' to the 'rule'?

Is 'Brazilian LD' even a correct term?  Or is "Lyme-like disease" more appropriate? Sorry, but IS IT, in fact Lyme disease?  Or a very similar illness by a not yet fully defined borrelia species???

Line 155: regarding use of BSK-II media - this was developed for North American borreliosis.  Given the vastly different ecology, animal hosts and tick-vectors in South America, it is possible that an entirely 'novel' culture media may need to be devised that MIGHT enable growth of novel borrelial species that may be causing Lyme-like illness in Brazilian.  Too bad Brazil does not have its own equivalent of an "NIAID Rocky Mountain Laboratory" to innovate and research novel culture media bearing in mind the unique aspects of South American ecology & tick & borrelial biolog.  Perhaps the authors could enlist the assitance of workers presently AT RML to try to solve this vexing problem (e.g. Patricia Rosa, Tom Schwan - retired I think)

Line 160:  "..became sick.  Therefore, we assumed...."

Line 172:  delete the hyphen before  Rhipicephalus

Line 187:   "...including capybaras....."

Line 200:  "....Amblyomma...

 Trying to classify BYS as Borrelia burgdorferi senso strictu seems like trying to "fit a 'square peg' in to a 'round hole'.  Considering the eons of evolutionary time that borreliae have been 'on the planet' and the vast differences in animal hosts and tick-vectors of disease between North and South America, it seems to the reviewer, that the likelihood of genetic identity between BYS and Lyme disease as defined by Bbss is vanishingly small and would 'defy' all odds that Bbss and BYS be found to be completely 'congruent'.  In fact, it crosses the reviewers mind whether there may be more correspondence between ticks vectors and borreliae between South America and Africa & Australia (e.g. Gondwana Land) than between South America and North America.  Pure speculation, of course.

Lines 301-302:  It is not necessary or appropriate for the authors to be 'self-deprecating' regarding the choice of the name BYS for the novel entity they observed and reported on.  BYS as a name, recognizes that, although there are many similarities to Lyme disease, there are also distinguishing features such that "Lyme disease" or even "Brazilian Lyme disease" would be a misleading designation.

Lines 344-346:  how dispositive is the presence of  the gene for flgE as 'proof' of Bbss as opposed to some other closely-related borrelia?  The reviewer is not a molecular biologist, but it would seem that carriage of one or even two genes with close homology to Bbss is likely not enough to make the claim of 'identity' with Bbss.  That would probably require whole genome sequencing. I think it would serve the authors better to remain somewhat agnostic as to the exact nature of the organism or organisms that are resulting in clinical BYS in Brazil, rather than making a premature claim to have proven that BYS is Lyme disease or due to Bbss.  Even making the claim of 'Lyme disease' may be too bold given the available data, the anomalous findings and the vast differences in vectors & hosts in Brazil and South America.  There is a certain judiciousness in 'reserving judgment' about the nature and identity of the etiologic agent or agents.  Even in California there has been shown to be a diversity of borreliae strains n Californian 'micro-environments'.  How much more opportunities for diversity in Brazil!!!   Fesler MC, Shah JS, Middelveen MJ, Du Cruz I, Burrascano JJ, Stricker RB. Lyme Disease: Diversity of Borrelia Species in California and Mexico Detected Using a Novel Immunoblot Assay. Healthcare (Basel). 2020 Apr 14;8(2):97. doi: 10.3390/healthcare8020097. PMID: 32295182; PMCID: PMC7349648.

Line 416:  "...for animal traction...." (what is meant by this term? Is it horses used for cattle herding?)

Line 445:  inability to isolate in culture remains a great challenge. enlisting assistance of scientists with interest and expertise in devising a novel culture-system, which surely could entail a major research effort, nonetheless would be worthwhile.

Lines 479-497:  endothelial is repeatedly mis-spelled.

Line 504:  "...serologic and molecular...."

Line 519:  "...,these camouflaged Borrelia burgdorferi sensu stricto...." Again, the reviewer is not a molecular biologist, but it seems that making the claim that BYS is due to Bbss seems not adequately supported by the evidence.  Perhaps some cases are due to Bbss (a few of the Case reports later in the mss. DID show fully diagnostic IgG WBs satisfying stringent CDC-criteria, but most of the BYS cases are unable to fulfill those epidemiologic case definition criteria.

Line 535:  "...activity permitting...."

Line 546:  "...non-phagocytic...."

Line 547:  "...cytopathic...."

Line 549:   "...as a strategy for ..." (delete comma after as)

Line 552:  "...travel to..." (only one 'l')

Line 592:  "...lia, allowing them..."  ( a comma, not a hyphen before allowing)

Line 618:  "...caused by Borrelia burgdorferi sensu stricto...." The reviewer does not think this has been adequately proven by the evidence presented (again, not a molecular biologist - but - it is felt a lot more proof of concordance or identity would be necessary to make this claim and to 'make it stick' without being severely challenged by other scientists.  Isolate the organism(s) in culture and/or do WGS on physically segregated spirochetal-like forms mechanically separated from fluids or culture media.

North American Lyme disease, as well as BYS can result in development of autoimmune phenomena, sometimes neuroautoimmunity and chronic fatigue-like states.

However, Schutzer  Fallon have accomplished proteomic studies of CSFs from Lyme patients and CFS patients and found them distinguishable Schutzer SE, Angel TE, Liu T, Schepmoes AA, Clauss TR, Adkins JN, Camp DG, Holland BK, Bergquist J, Coyle PK, Smith RD, Fallon BA, Natelson BH. Distinct cerebrospinal fluid proteomes differentiate post-treatment lyme disease from chronic fatigue syndrome. PLoS One. 2011 Feb 23;6(2):e17287. doi: 10.1371/journal.pone.0017287. PMID: 21383843; PMCID: PMC3044169.

Line 623:  "...confusion..." (not confusions)

Line 638:  "....attempting..."

Line 651:  "...may elapse..."

Line 659:  The spirochete also may invade directly through tissue planes and not only by lymphatic and bloodstream.  Klempner MS, Noring R, Epstein MP, McCloud B, Hu R, Limentani SA, Rogers RA. Binding of human plasminogen and urokinase-type plasminogen activator to the Lyme disease spirochete, Borrelia burgdorferi. J Infect Dis. 1995 May;171(5):1258-65. doi: 10.1093/infdis/171.5.1258. PMID: 7751701.  See also Straubinger's Beagle dog studies RE: centripetal spread of borreliae Straubinger RK. PCR-Based quantification of Borrelia burgdorferi organisms in canine tissues over a 500-Day postinfection period. J Clin Microbiol. 2000 Jun;38(6):2191-9. doi: 10.1128/JCM.38.6.2191-2199.2000. PMID: 10834975; PMCID: PMC86761.

Line 671:  "...antigens.." (not antigenes)

Lines 675-676:  In contrast to or addition to PTLDS, citation of Shor et al regarding chronic Lyme disease would seem appropriate Shor S, Green C, Szantyr B, Phillips S, Liegner K, Burrascano JJ Jr, Bransfield R, Maloney EL. Chronic Lyme Disease: An Evidence-Based Definition by the ILADS Working Group. Antibiotics (Basel). 2019 Dec 16;8(4):269. doi: 10.3390/antibiotics8040269. PMID: 31888310; PMCID: PMC6963229.

Line 688:  "...certain foods..." Consideration of development of Mast Cell Activation Syndrome might be mentioned as some practitioners dealing with Lyme disease have felt that to be an occasional feature with multiple allergic reactions Theoharides TC, Kavalioti M, Martinotti R. Factors adversely influencing neurodevelopment. J Biol Regul Homeost Agents. 2019 Nov-Dec;33(6):1663-1667. doi: 10.23812/19-33n6Edit_Theoharides. PMID: 31928596.

Line 706: "...7.1.2. Early disseminated Infectious Stage..." The reviewer wonders, considering the already advanced nature of some of the manifestations described in the section, whether "Early disseminated..." is the most appropriate designation, because many suggest a pathologic process already well-advanced.

Line 710:  "...annular..." (not anular)

Line 710-7ll:  "..lymphocytoma.."

Line 725:  "..Typically,  ...."

Lines 737-738:  "....lymphomononuclear meningitis...."

Line 741:  use of the designation B. burgdorferi in this line may be questionable and the reviewer wonders whether or not "...intrathecal antibodies to BYS borreliosis..." might not be more suitable, unless there is 'bullet-proof' evidence of B. burgdorferi antibodies per se.

Line 743:  "...pattern of sensory and/or motor...."

Lines 748-749:  Are the authors saying that 20% of all neuropsychiatric cases in the nation of Brazil are linked to BYS borreliosis?  If that is the intent, it might be good to clarify what is meant.

Line 763:  "...often of dermatomal distribution.." (is that what is meant by the term "metameric"??)

Line 750:  "....psychotic...."

Line 754:  "....myelopathies..."

Line 763:  "....CSF analysis..." (not CFS analysis)

Line 764:   "...and identification..."  (not "dentification")

Lines 738-743 and lines 763-765 are somewhat redundant.  Perhaps this point need be made only once (e.g. the importance of CSF analysis in cases with neurologic and/or neuropsychiatric features).

Line 772:  "...ongoing infection...." (not infectio)

Line 777:  "...biological therapy...."  It might be good to clarify what the authors mean by the term.  Are they talking about biologic modifiers such as etanercept, adalimumab or rituxan??  Some clarification would be good.

Line 778:  "...morphology of B. burgdorferi..." Sorry to keep 'beating this drum' but would not "..morphology of BYS borreliae..." be more accurate and supportable?

Line 784:   "...Bannwarth syndrome..." (not Bannwart)

Line 788:  "...atrio-ventricular conduction system..." (rather than the awkward " atrio-ventricular electrical stimulus conduction...")

Line 789: In North American Lyme borreliosis when the AV conduction system is affected, antibiotic therapy and at least temporary pacemaker placement is not uncommonly required.  So, if that is NOT the case with BYS borreliosis, well, that is distinguishing between the two 'entities'.

Line 792:  "....of the oculomotor system, causing.."

Lines 796-828:  The question of BYS borrelial persistence, which the authors are observing in their case material might be well-served by incorporating here or elsewhere in the mss. or conclusions, a review and discussion of the now ample evidence of persistence of B. burgdoreri in the worldwide peer-reviewed literature my many authors, many independent scientific research groups.  Much of that literature is cited in the Shor article.  (Shor S, Green C, Szantyr B, Phillips S, Liegner K, Burrascano JJ Jr, Bransfield R, Maloney EL. Chronic Lyme Disease: An Evidence-Based Definition by the ILADS Working Group. Antibiotics (Basel). 2019 Dec 16;8(4):269. doi: 10.3390/antibiotics8040269. PMID: 31888310; PMCID: PMC6963229.) This has been demonstrated despite application of antibiotics, and sometimes prolonged treatment.  This is a problematic feature of the borrelioses and requires application of more research to devising improved methods of treatment.

Regarding development of autoimmunity, if not already included, some discussion of molecular mimicry as a mechanism would be appropriate. Steere AC, Dwyer E, Winchester R. Association of chronic Lyme arthritis with HLA-DR4 and HLA-DR2 alleles. N Engl J Med. 1990 Jul 26;323(4):219-23. doi: 10.1056/NEJM199007263230402. Erratum in: N Engl J Med 1991 Jan 10;324(2):129. PMID: 2078208.

Lines 830 and following: This would be a good area to raise in a more explicit way, the possible role of tick- and vector-borne co-infections that (at least in North America) often are found in association with borrelial infection including piroplasmoses, bartonellosis and miscellaneous infectious agents.  When identified and treated, sometimes the 'chronic fatigue' states remit.  This may require combination therapies to 'cover' infectious agents that may not respond adequately to therapy aimed ONLY at a borreliosis. The work of Breitschwerdt and co-workers, which at this point is very substantial and which demonstrates a high degree of overlap in symptoms of borreliosis and bartonellosis is pertinent. For example see: Maggi RG, Mozayeni BR, Pultorak EL, Hegarty BC, Bradley JM, Correa M, Breitschwerdt EB. Bartonella spp. bacteremia and rheumatic symptoms in patients from Lyme disease-endemic region. Emerg Infect Dis. 2012 May;18(5):783-91. doi: 10.3201/eid1805.111366. PMID: 22516098; PMCID: PMC3358077. as well as their prodigious publications.

Line 871:  "...onset of drug or food allergies" (again, consideration of triggering of Mast Cell Activation Syndrome in pre-disposed individuals may merit discussion here.

Line 894:  "...corticotrophin..." 

Line 916:  "...arthritis.." (not artrhitis)

Line 925: "STPE" why is the abbreviation STPET used? What does the 'terminal' T signify.  STPET is also in Figure 18. Skin Tissue Protein Extract (STPE) so why is it listed as STPET????

Lines 932 & following:  Citation of the work of Alaedini and colleagues regarding neuroautoimmunity for Lyme disease might be pertinent to include Alaedini A, Latov N. Antibodies against OspA epitopes of Borrelia burgdorferi cross-react with neural tissue. J Neuroimmunol. 2005 Feb;159(1-2):192-5. doi: 10.1016/j.jneuroim.2004.10.014. Epub 2004 Nov 26. PMID: 15652419.

Line 965:  "...doxycycline..."

Line 966:  "...lumbar..."

Lines 968-972:  Many clinicians have found that hyper-extended antimicrobial treatment, considering borreliae may not be cured with currently available treatment methodologies, can improve the clinical conditions of persons with even advanced neurologic and neuropsychiatric (Lyme) borreliosis.  This is in contradistinction to the prevailing view that long-term or even 'open-ended' antibiotic therapy is useless.  Of course, this question remains one that is highly controversial.  Again, refer to Shor et al. Also, the extensive writings of psychiatrists Fallon as well as Bransfield speak to these issues. Fallon BA, Nields JA, Burrascano JJ, Liegner K, DelBene D, Liebowitz MR. The neuropsychiatric manifestations of Lyme borreliosis. Psychiatr Q. 1992 Spring;63(1):95-117. doi: 10.1007/BF01064684. PMID: 1438607.   Bransfield RC. Neuropsychiatric Lyme Borreliosis: An Overview with a Focus on a Specialty Psychiatrist's Clinical Practice. Healthcare (Basel). 2018 Aug 25;6(3):104. doi: 10.3390/healthcare6030104. PMID: 30149626; PMCID: PMC6165408.

Line 1004:  "....October..."

Line 1011: "..temper tantrums.." (not tempers)

Line 1012:  "....low grade fever..."

Lines 1021-1022:  "...serologic tests for B. burgdorferi were positive.." (specify just what kind of serologic test was positive? What were the 'criteria' of 'positivity'?

Line 1035:  "...during the course of her illness, she is..."

Citation of the works of Ying Zhang, Jayakumar Rajadas, Kim Lewis, John Scott, Conrad & Persing, Breitschwerdt, all speaking to the limitations of customary treatment approaches, need for improved methods of therapy, the other tick- and vector-borne infections that may accompany the borrelioses, and the still novel agents that are being discovered (e.g. the complexity of Nature - and even more so in South America where MUCH remains to be discovered) is pertinent and would add depth to the current mss. IF the authors are willing to take this more expansive literature ON. Likely, though, this would require extensive revision and would delay publication of the mss.  (Scott JD, Sajid MS, Pascoe EL, Foley JE. Detection of Babesia odocoilei in Humans with Babesiosis Symptoms. Diagnostics (Basel). 2021 May 25;11(6):947. doi: 10.3390/diagnostics11060947. PMID: 34070625; PMCID: PMC8228967. & Feng J, Wang T, Shi W, Zhang S, Sullivan D, Auwaerter PG, Zhang Y. Identification of novel activity against Borrelia burgdorferi persisters using an FDA approved drug library. Emerg Microbes Infect. 2014 Jul;3(7):e49. doi: 10.1038/emi.2014.53. Epub 2014 Jul 2. PMID: 26038747; PMCID: PMC4126181. & Pothineni VR, Wagh D, Babar MM, Inayathullah M, Solow-Cordero D, Kim KM, Samineni AV, Parekh MB, Tayebi L, Rajadas J. Identification of new drug candidates against Borrelia burgdorferi using high-throughput screening. Drug Des Devel Ther. 2016 Apr 1;10:1307-22. doi: 10.2147/DDDT.S101486. PMID: 27103785; PMCID: PMC4827596. & Liegner KB. Disulfiram (Tetraethylthiuram Disulfide) in the Treatment of Lyme Disease and Babesiosis: Report of Experience in Three Cases. Antibiotics (Basel). 2019 May 30;8(2):72. doi: 10.3390/antibiotics8020072. PMID: 31151194; PMCID: PMC6627205. & Leimer N, Wu X, Imai Y, Morrissette M, Pitt N, Favre-Godal Q, Iinishi A, Jain S, Caboni M, Leus IV, Bonifay V, Niles S, Bargabos R, Ghiglieri M, Corsetti R, Krumpoch M, Fox G, Son S, Klepacki D, Polikanov YS, Freliech CA, McCarthy JE, Edmondson DG, Norris SJ, D'Onofrio A, Hu LT, Zgurskaya HI, Lewis K. A selective antibiotic for Lyme disease. Cell. 2021 Oct 14;184(21):5405-5418.e16. doi: 10.1016/j.cell.2021.09.011. Epub 2021 Oct 6. PMID: 34619078; PMCID: PMC8526400. & Persing DH, Conrad PA. Babesiosis: new insights from phylogenetic analysis. Infect Agents Dis. 1995 Dec;4(4):182-95. PMID: 8665084.

Lines 1055-1084: surely neuro-autoimmune mechanisms may be operative which require immune-modulating interventions, however it would appear, from a broad perspective, that very extended antimicrobial treatments have not been systematically applied to these advanced cases which MIGHT benefit from such an approach.  This notion is 'heretical' according to the CDC and the IDSA.  Nonetheless, this has been the experience of many clinicians and patients dealing with advanced chronic and neurologic and neuropsychiatric Lyme borreliosis in North America.

The reviewer feels privileged to review this manuscript and wishes to compliment the authors for their truly pioneering work in describing and studying what surely appears to be a South American correlate of Lyme borreliosis (BYS borreliosis) which still has many puzzling aspects and is still incompletely characterized and merits much further study including novel and cutting-edge scientific studies.  The Reviewer suggests that the authors avoid 'forcing' their 'square peg' of BYS borreliosis in to the 'round hole' of Lyme borreliosis (e.g. Bbss) although some of the cases may actually qualify as such...but many may not.  A more agnostic position, reserving judgment and calling for more intensified research with application of more financial and scientific resources, would, the Reviewer believes, enhance and strengthen the position of the authors, rather than prematurely concluding that BYS IS due to Bbss, which the reviewer feels the data is not adequate to justify.

Nonetheless, this is the AUTHORS' paper, and not the Reviewers, who is only making suggestions, which are intended to be constructive.  Just how the authors choose to finalize their mss. and to what extent (or not) they wish to revise it (other than obvious English language usages and syntax corrections) is up to the Authors and the Editors to jointly decide.  The authors are presenting novel observations that are worthy in and of themselves and have already done extensive work, bringing to light the intriguing BYS which indeed shares many clinical similarities to North American Lyme borreliosis due to Bbss and Bbsl.

Author Response

Comments made by the reviewer have been all valuable and very helpful for revising and improving our paper. We have studied comments carefully and have made correction which we hope meet with approval. According to the large amount of revisions, point by point response has not been included hereafter c. Anyway, all the revised portions are marked in red in the manuscript.